# Robotic-OCT guided inspection and microsurgery of monolithic storage devices

Bin He[1,2,4], Yuxin Zhang[1,2,4], Lu Zhao[1,4], Zhenwen Sun[1], Xiyuan Hu[3], Yanrong Kang[1], Lei Wang[1], Zhihui Li[1], Wei Huang[1], Zhigang Li[1], Guidong Xing[1], Feng Hua[1], Chengming Wang[2], Ping Xue ®[2] ✉ & Ning Zhang ®[1] ✉

Data recovery from monolithic storage devices (MSDs) is in high demand for legal or business purposes. However, the conventional data recovery methods are destructive, complicated, and time-consuming. We develop a robotic-arm-assisted optical coherence tomography (robotic-OCT) for non-destructive inspection of MSDs, offering ~7 μm lateral resolution, ~4 μm axial resolution and an adjustable field-of-view to accommodate various MSD sizes. Using a continuous scanning strategy, robotic-OCT achieves automated volumetric imaging of a micro-SD card in ~37 seconds, significantly faster than the traditional stop-and-stare scanning that typically takes tens of minutes. We also demonstrate the robotic-OCT-guided laser ablation as a microsurgical tool for targeted area removal with precision of ±10 μm and accuracy of ~50 μm, eliminating the need to remove the entire insulating layer and operator intervention, thus greatly improving the data recovery efficiency. This work has diverse potential applications in digital forensics, failure analysis, materials testing, and quality control.

Monolithic storage devices (MSDs), such as micro-SD, USB flash drive, compact flash, and multimedia card (MMC), are popular for storing data in various electronic devices and systems because of their small size, high performance, reliability, and ease of use[1]. Data recovery from MSDs is in high demand for legal or business purposes, as it helps individuals and organizations retrieve critical or sensitive information that has been lost due to damage, malfunction or deliberate deletion.

However, data recovery from MSDs remains a major challenge for forensic investigations or data recovery services[2]. MSDs have an integrated design where the interface, controller, and flash memory chips are on a printed circuit board (PCB) that is covered by an opaque insulating layer. Therefore, current methods require manually sanding off the entire insulating layer to expose the underlying PCB. Then, the technological pins that access the flash memory can be identified and used to recover the data directly from the memory without going through the device's interface and controller.

The current methods suffer from several major limitations. First, manual removal of the entire insulating layer may damage the copper layers or other components on the PCB, affecting the device's integrity and hampering the data recovery[3]. This is especially important in legal proceedings, where the physical evidence must be intact to ensure the reliability and admissibility of the recovered data. Another limitation is the difficulty to control the location and depth of removal. Manual sanding uses abrasive materials such as sandpaper to scrape off the entire insulating layer from the MSD surface, which is challenging to control without damaging the underlying components or to expose only the targeted pins. Sanding can also create dust and debris that can interfere with the recovery process. Thirdly, the removal process is time-consuming, labor-intensive and requires specialized skills, tools, and expertise, which significantly increases the workload and complexity.

[1]Institute of Forensic Science, Ministry of Public Security, 100038 Beijing, China. [2]State Key Laboratory of Low-dimensional Quantum Physics and Department of Physics, Tsinghua University and Beijing Advanced Innovation Center for Structural Biology, 100084 Beijing, China. [3]School of Computer Science and Engineering, Nanjing University of Science and Technology, 210094 Nanjing, China. [4]These authors contributed equally: Bin He, Yuxin Zhang, Lu Zhao. ✉e-mail: xuep@tsinghua.edu.cn; zhangning@cifs.gov.cn

Some researchers have suggested using X-ray radiography as a nondestructive method to examine the internal structure of a MSD, eliminating the need to erase the insulating layer[4,5]. However, this method has some limitations as well. The two-dimensional (2D) X-ray image overlays all the views from the top to the bottom of the sample, which makes it impossible to distinguish the elements and their layers. The micro-computed tomography (μCT) can produce a three-dimensional (3D) X-ray image, but the scanning and reconstruction procedure are time-consuming[6–8]. Most importantly, X-rays can be harmful to operators and may introduce irreparable bit errors to stored data due to charge loss and trapping[9,10]. Therefore, there is a need for techniques that can inspect MSDs accurately and rapidly without harming the operator, the device, or the stored data, and improve the efficiency and reliability of data recovery.

Optical coherence tomography (OCT) offers a promising approach to address the above limitations. OCT is considered as an optical biopsy based on low-coherence interferometry that can provide non-contact and nondestructive imaging of MSDs' internal microstructures. This helps examiners inspect and diagnose the device without removing the insulating layer. Importantly, OCT uses low-power near-infrared light instead of X-rays, avoiding hazardous radiation or bit error problems that can harm operators and stored data. OCT can also perform quantitative tomographic analysis for reverse engineering and defect analysis. Moreover, OCT can obtain depth-resolved information and detailed 3D volumetric images in a very fast manner[11]. While OCT has traditionally been widely used in the biomedical field, and other non-biomedical fields such as art conservation[12], industrial inspection[13], and forensic science[14–18], there have been no studies aimed at inspecting MSDs for the purpose of data recovery.

When applying OCT to MSD inspection, several challenges arise, including sample sizes, surface reflectivity, and scanning strategy. First, MSDs come in various types and sizes, ranging from small micro-SD cards to larger MMCs. Conventional OCT systems have a fixed and limited field-of-view (FOV), making it difficult to adapt to large or complex objects[19]. Moreover, highly reflective surfaces within MSD structures can cause saturation signals and artifacts, hindering the visualization of underlying structures. Optimizing the positions and angles of the OCT scanner can help mitigate direct reflections and improve image quality, but it is challenging to maintain high repeatability and consistency between scans. Furthermore, previous studies often rely on a stop-and-stare scanning strategy to extend the FOV[20–23], where the scanner pauses at specific locations to capture data before moving to the next position. This approach is straightforward and easy to implement. However, it can be time-consuming, inefficient, and result in inconsistent image acquisition due to frequent stopping and repositioning, especially when dealing with larger or irregularly shaped objects.

Combining robotics with OCT offers innovative solutions to overcome these challenges, providing benefits including flexible and accurate imaging, process automation, and consistent image acquisition. The robotic arm can be programmed to move the scanner precisely and automatically to different positions and angles, which allows for an adjustable FOV to accommodate various sample sizes and shapes while reducing saturation signals and artifacts. To overcome the limitations of the traditional stop-and-stare scanning approach with a robotic arm, a continuous scanning strategy can be implemented, which enables the scanner to acquire data continuously while in motion. By eliminating interruptions or discontinuities in the imaging process, the continuous scanning strategy can significantly reduce the overall scanning time and ensure a uniform and seamless image acquisition across the entire area of interest. Inspired by image-guided surgical interventions in the medical field, a microsurgical tool can also be developed for MSDs by integrating OCT imaging, robotic

arm and laser ablation capabilities. With the aid of OCT images as guidance, the robotic arm can accurately direct the laser to specific locations, enabling selective removal of targeted areas or structures. This approach can minimize damage to the device and facilitate data recovery by eliminating the need to remove the entire insulating layer and reducing operator intervention.

In this study, we reported the exploration of robotic-arm-assisted OCT (robotic-OCT) for nondestructive inspection and microsurgery of several common types of MSDs. We integrated a robotic arm into a spectral-domain OCT system, and obtained 3D volumetric, 2D cross-sectional, and en face images of the devices by using robotic positioning and a continuous scanning strategy. This enabled us to achieve nondestructive, high-resolution, and automated imaging with large and adjustable FOV in a fast and flexible manner. The internal PCB traces were clearly revealed, with the relevant technological pins for data recovery and the distribution of vias accurately identified, providing important information for reverse engineering. In addition, the insulating layer in different samples was characterized by cross-sectional analysis to provide quantitative information for controlling the removal depth. Furthermore, we examined and diagnosed three damaged micro-SD cards (scratched, cracked and burned), using en face images to detect different types of defects. Finally, robotic-OCT-guided laser ablation was performed to automatically and precisely remove specific areas of the insulating layer covering the desired pins, allowing safe and effective access to the flash memory with minimal damage to the device. The results demonstrate robotic-OCT as a powerful technique for inspecting and performing microsurgery on MSDs, which offers a significant improvement over conventional methods and increases data recovery efficiency.

## Results

### Robotic-OCT imaging with continuous scanning strategy

We developed a custom-built robotic-OCT system (see "Methods" for details) that utilized an OCT probe with a galvo scanner for fast-axis lateral scanning, and a robotic arm for slow-axis lateral scanning, as illustrated in Fig. 1a, b. We imaged a micro-SD card as an example using this system and implemented a continuous scanning strategy for robotic-OCT imaging. The scanning trajectory of the robotic arm for the micro-SD card was indicated by the green arrows in Fig. 1c. The scanning speed of the robotic arm was ~1 mm/s. The robotic arm first moved continuously along the slow axis from point A to point B in 15.5 s, while the OCT probe constantly acquired 3000 B-scans in real time with a lateral scan range of ~7 mm. These B-scans were stacked to create a 3D OCT volume data of the scanned area (Region 1, 3000 × 1000 × 2048 voxels), and the corresponding en face image (left in Fig. 1d) at the depth of the PCB within the sample was obtained by slicing through the 3D volume data. The robotic arm then moved from point B to point C in 6 s, reversed the scanning direction and performed another continuous scanning of the adjacent area (Region 2, right in Fig. 1d) along the slow axis from point C to point D in 15.5 s with the same dimensions as Region 1. As a result, the FOV was extended to 11.3 mm × 15.5 mm by automatically fusing the en face images of Region 1 and Region 2 with a 10% overlap to create the image of the internal PCB trace encompassing the entire micro-SD card, as shown in Fig. 1e. This strategy could be further conducted to produce a seamless large image of any desired area of interest by automatically fusing the en face images of a series of adjacent regions scanned by the robotic arm. The experiment demonstrated that the robotic-OCT with a continuous scanning strategy enabled automated, flexible and fast imaging to cover an adjustable FOV.

### Tomographic analysis of the multilayer structure

The 3D OCT images in Fig. 2a, b provided two different detailed views of the multilayer structure of part of a micro-SD card (Card 1), which included the plastic housing, PCB and insulation layers as illustrated in

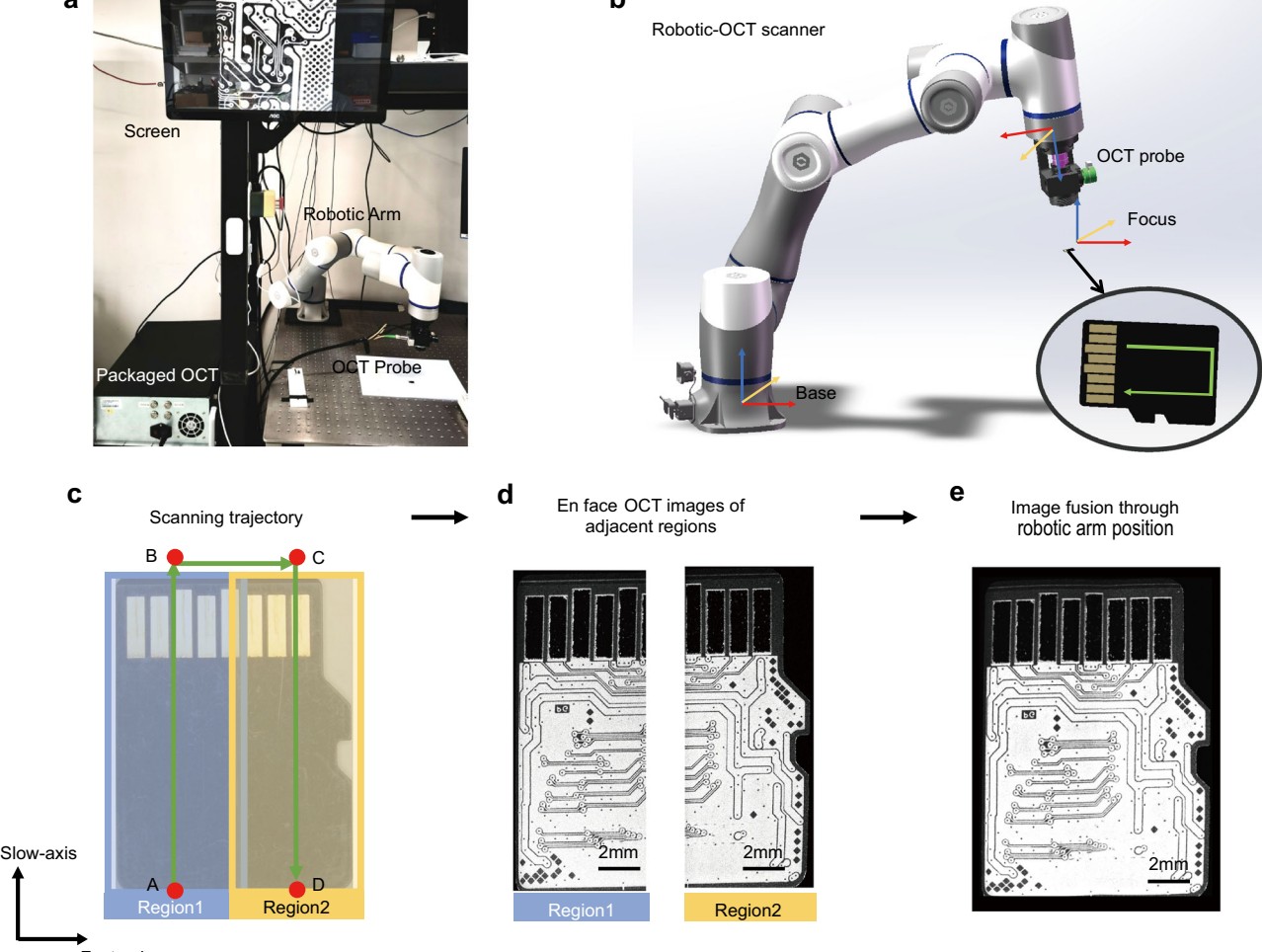

**Fig. 1 | Illustration of the continuous scanning robotic-OCT system.**
**a** Photograph of the robotic-OCT system. **b** Schematic of the robotic-OCT scanner. The green arrow line indicates the scanning trajectory of the robotic arm. **c** Scanning trajectory of the robotic arm represented by the green arrow line, and the two adjacent continuous scanning regions, represented by blue (Region 1) and yellow (Region 2); the robotic arm moves along the slow axis from point A to point B in 15.5 s, moves from point B to point C in 6 s, reversed the scanning direction and moves along the slow axis from point C to point D in 15.5 s. **d** The en face OCT images of Region 1 and Region 2. **e** The fused image covering the entire area of a micro-SD card.

Fig. 2c. En face OCT images at different depths were extracted from the 3D volume data to reveal specific features of the card, as shown in Fig. 2d. The top image in Fig. 2d shows the surface of the pinout side of the micro-SD card, displaying the screen-printed text of the model information. The middle image provided a clear view of the internal PCB trace on the pinout side, showing the network of wiring, copper, insulation, and contacts that were not obscured by the surface text. In contrast, the bottom image presented the PCB trace on the dice side, which was not as clear as that on the pinout side due to the light reflection from the copper layer on the pinout side. This observation was supported by the fact that the bright regions in the middle image corresponded to the dark areas in the bottom image.

In Fig. 2e, we presented averaged A-scans ($n = 10$) obtained from two distinct positions, labeled as P1 and P2. These positions were identified and marked using yellow rectangles in Fig. 2d. By analyzing the A-scans, we made several important observations. First, both A-scans exhibited a clear first peak (S1), which corresponded to the surface of the micro-SD card, indicating the starting point of the penetration depth. Moving further into the card, the A-scan from position P1 displayed a second peak (S2), representing the boundary between the insulating layer and the PCB. This peak resulted from the reflection of the incident light by the copper conductive layer on the

pinout side of the PCB. By calculating the optical path length (OPL) between S1 and S2, we estimated the thickness of the insulating layer to be ~23 µm (see "Methods" for details). In addition, the A-scan from position P2 exhibited a second peak (S3), indicating the reflection of light from the copper conductive layer on the dice side of the PCB. Notably, there was no corresponding copper conductive layer at the same position on the pinout side, as evident from the absence of a reflected signal. By calculating the OPL between S2 and S3, we estimated the thickness of the PCB to be ~250 µm. Consequently, the total penetration depth achieved by our OCT system into this sample was determined to be ~273 µm.

**Visualization of internal PCB traces and characterization of the insulating layer**
In Fig. 3, we presented representative OCT imaging results of four micro-SD card samples (Card 1, 2, 3, 4, see "Methods" for details), each with distinct brands, models, surface roughness, and insulating layer thicknesses. Card 1 was a brand-new card obtained directly from the manufacturer. Card 2 and Card 3 were two used cards with distinct surface roughness obtained from the second-hand markets. Card 4 was seized from a real case without any brand information. Figure 3a, b shows the photographs and the en face OCT images of these four cards

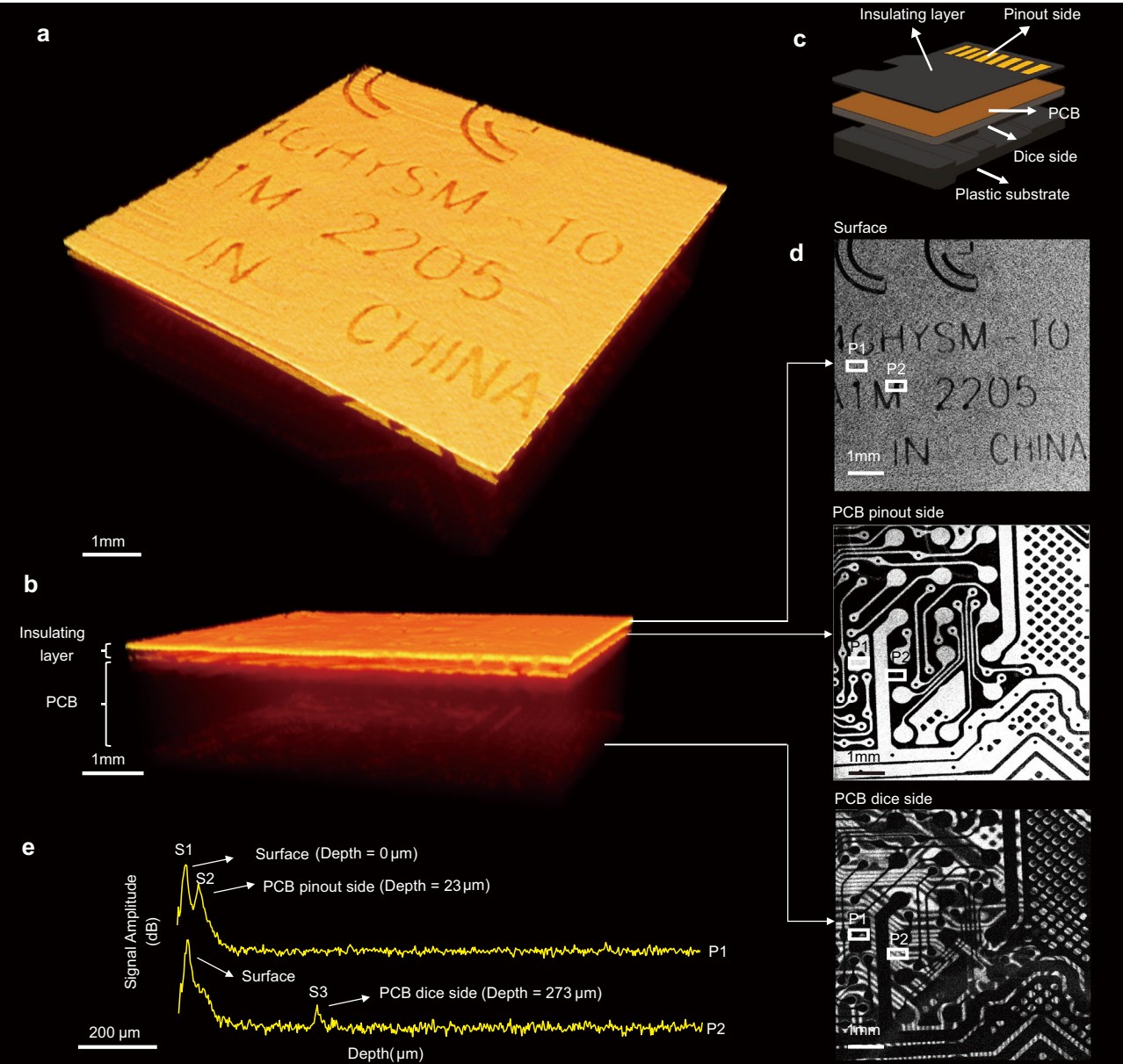

**Fig. 2 | Tomographic analysis of the multilayer structure of a micro-SD Card.** **a** 3D OCT image of the multilayer structure in a top-to-bottom view. **b** 3D OCT image of the multilayer structure in a cross-sectional view. **c** Diagram of the multilayer structure of a micro-SD card, including an insulating layer, PCB and plastic substrate. **d** En face OCT images at three different depths: Top−surface (depth = 0 μm), Middle −PCB pinout side (depth = 23 μm), Bottom−PCB dice side (depth = 273 μm). **e** Averaged A-scans at two different positions, P1 and P2, marked as white rectangles in (**d**). S1 corresponds to the surface of the micro-SD card. S2 represents the boundary between the insulating layer and the PCB. The distance between S1 and S2 represents the thickness of the insulating layer, estimated to be ~23 μm. S3 indicates the reflection of light from the copper conductive layer on the dice side of the PCB, suggesting the thickness of the PCB to be ~250 μm.

from the pinout side, where we can clearly visualize the internal PCB traces hidden beneath the insulating layer on the pinout side regardless of the presence of screen-printed text or insulating layers. The corresponding B-scans at the red dashed lines in Fig. 3b are also shown in Fig. 3c, where the different layers can be clearly distinguished. These B-scans clearly showed the presence of a black region within the insulating layer, indicating that no light was scattered from the inside of the layer. We conducted OCT imaging on a diverse set of over 80 micro-SD cards, including different brands and models collected from multiple sources. Our observations consistently revealed that the insulating layers of these cards exhibited transparency to the near-infrared light used in our OCT systems. These results demonstrated that the material of the insulating layer did not significantly affect the OCT inspection process.

We obtained the thickness of the insulating layer of micro-SD cards from the optical path length (OPL) between first two boundaries in B-scan, as OPL equals product of group refractive index of insulating layer (~1.5) and its geometric thickness. The standard deviation (STD) of the layer thickness was also measured to estimate variability and uniformity of the layer, as shown in Table 1. The results showed that Card 3 had a maximum thickness of 36.2 μm, while Card 1 had a relatively low thickness of 21.6 μm. Card 2, 3, and 4 had higher STD (2.9 μm, 2.5 μm, 2.9 μm) than Card 1 (1.7 μm) due to wear and tear of the used cards. Analyzing the properties of the insulating layer using OCT could provide additional information about the device fabrication process and determine whether the insulating layer has been removed to the desired depth. Any deviations from desired removal depth could lead to defects or failures in the circuit. These results

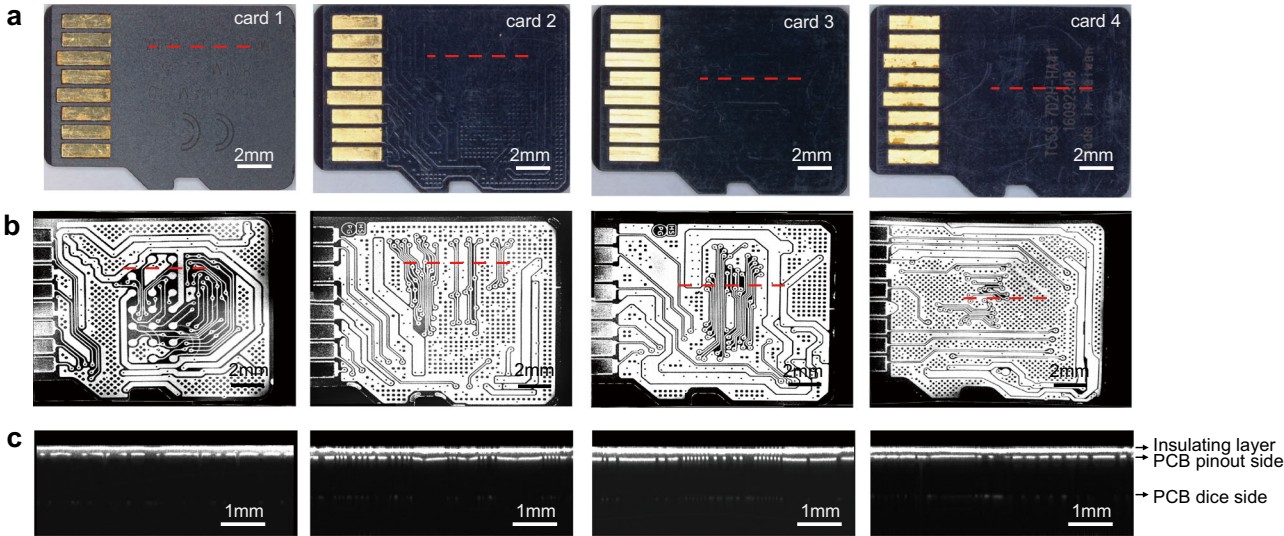

**Fig. 3 | Visualization of the internal PCB traces of different micro-SD cards. a** Photographs of four micro-SD cards on their pinout side. **b** The corresponding en face OCT images. **c** The B-scans at the positions indicated by the red dashed lines in (**a**, **b**), clearly show the insulating layer, PCB pinout side, and PCB dice side.

suggested that OCT imaging was a valuable tool not only for non-destructively revealing PCB traces but also for quantitatively characterizing the insulating layer which was favorable for accurately controlling and evaluating the removal process.

### Identification of pins and vias

We manually removed the insulating layer from Card 4 using sandpaper as in the traditional method, and obtained a microscopic image of the exposed PCB on the pinout side, as shown in Fig. 4a. We then compared it with the corresponding en face image obtained by robotic-OCT imaging before the removal process, as shown in Fig. 4b). We assigned different colors to the identified tracks in Fig. 4b and it was found that the OCT image was consistent with the microscope image. The PCB layout was retrieved from the existing micro-SD card database, and two groups of technological pins were identified: D0, D1, D2, D3, D4, D5, D6, and D7, which represented the data I/O signal pins, and address latch enable (ALE), chip enable (CE), command latch enable (CLE), read enable (RE) and write enable (WE), which denoted the command signal pins. These assigned pin names were marked in Fig. 4a, b. Using these pins, an examiner could extract data directly from the flash memory chip by using an adapter and a data recovery system as shown in Fig. 4c.

In addition to pin identification, the OCT image also allowed for clear visualization of the vias on the PCB, which are the holes used for electrical connections between different PCB layers. These vias appeared as white spots in the en face OCT image at -66 μm below the surface of the sample, as shown in Fig. 4d. The choice of imaging at this specific depth was made to align with the middle portion of the PCB, effectively avoiding potential interference from PCB wiring and solid soldering holes, and ensuring clear visualization of the vias. The size of the vias was estimated to be -14 μm in diameter, which was easily

detected by the high-resolution OCT imaging. This experiment demonstrated the practical application of OCT imaging to identify the pins and vias without the need to expose the PCB, and to provide critical information for the examiners to reverse engineer the target device.

### Diagnosis of damaged micro-SD cards

We simulated three common types of damage to micro-SD cards (scratched, cracked, and burned) as shown in Fig. 5c, g, k, and evaluated the effectiveness of the proposed robotic-OCT's ability to detect and diagnose these defects. For the scratched sample (Card 5), the en face OCT image revealed two scratches of different depths, as shown in Fig. 5b, which was the enlarged image of the area within the red box in Fig. 5a. It was observed that the severe scratch led to cutting of the tracks and could affect the connection with a card reader, whereas the minor scratch did not affect the integrity of the tracks. The different scratch depths could also be evaluated by the B-scan, as shown in Fig. 5d, which was captured at the red dashed line in Fig. 5b. The minor scratch was found to have a depth of -14 μm, which is smaller than the thickness of the insulating layer (-30 μm). This indicated that the scratch did not cause any damage to the pinout side of the PCB. On the other hand, the severe scratch had a depth of -60 μm, exceeding the thickness of the insulating layer and suggesting a potential risk of damaging the pinout side of the PCB. However, considering the overall PCB thickness of -250 μm, we can confidently conclude that the scratch did not reach the dice side and did not impact the flash die.

For the cracked sample (Card 6), the cracked trace was visible in the en face OCT image, as shown in Fig. 5e, f, and the corresponding B-scan, as shown in Fig. 5h. Although it rendered the micro-SD card unusable, our examination of en face OCT images revealed that the fractures did not affect the core pins necessary for data recovery. This finding indicated that there was still a possibility that data recovery procedures could be performed on the intact region.

For the burned sample (Card 7), some areas of the en face OCT image exhibited relatively low brightness due to the enhanced absorption of light in the burned area, which weakened the OCT signal, as shown in Fig. 5i. We could identify the burnt areas through the B-scans, as illustrated by the red boxed area in Fig. 5l. By thoroughly analyzing all the B-scans, we could delineate the boundary of the burnt area in the en face OCT image as shown in Fig. 5j. These results suggested that our method could effectively detect and evaluate a variety

**Table 1 | The thickness and STD of the insulating layers in different samples**

|  | Thickness (μm) | STD (μm) |
|---|---|---|
| Card 1 | 21.6 | 1.7 |
| Card 2 | 31.2 | 2.9 |
| Card 3 | 36.2 | 2.5 |
| Card 4 | 31.1 | 2.9 |

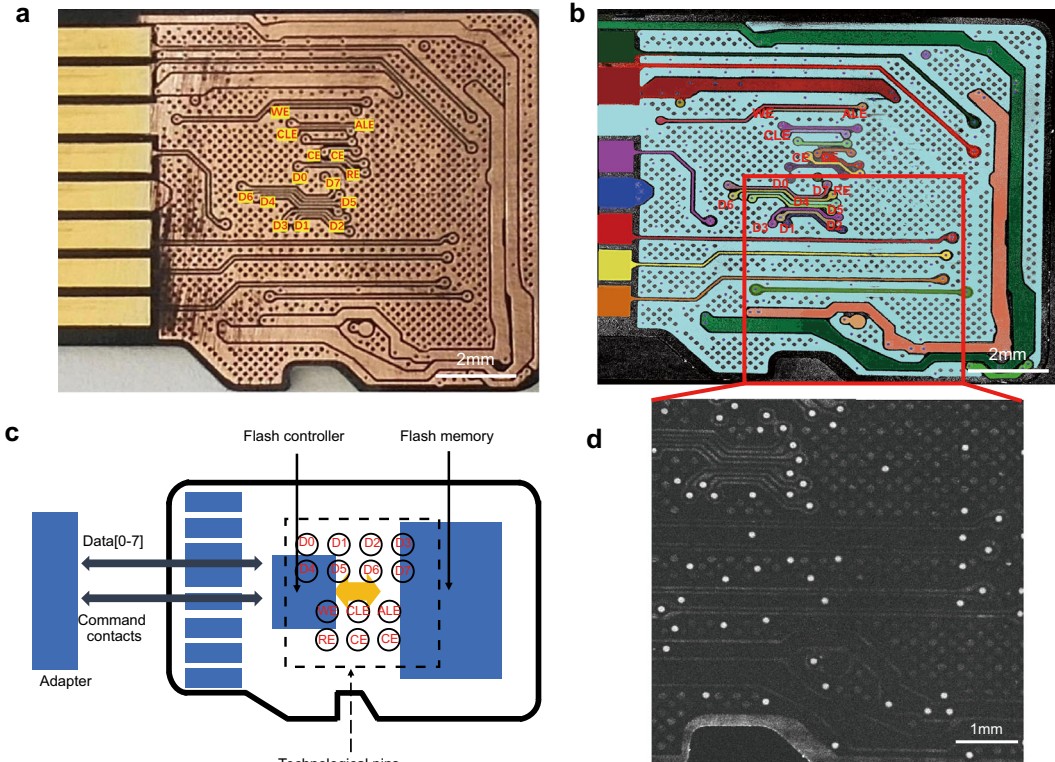

**Fig. 4 | Identification of pins and vias. a** The microscopic image of the exposed PCB on the pinout side obtained after removing the insulating layer. **b** The en face image of the PCB on the pinout side obtained by robotic-OCT imaging before the removal process. **c** The scheme of data extraction from flash memory of a micro-SD card. **d** The distribution of vias appearing as white spots in the en face image of the region indicated by the red rectangle in (**b**), at a depth of ~66 μm below the surface of the insulating layer. D0, D1, D2, D3, D4, D5, D6, and D7 represent the data I/O signal pins, while address latch enable (ALE), chip enable (CE), command latch enable (CLE), read enable (RE) and write enable (WE) denote the command signal pins.

of defects such as the cutting traces, scratches, cracked traces, burned areas and broken connections, for repairing or recovering data from the damaged devices, which could help determine whether the card was repairable or needed to be replaced, and provide valuable information for the following data recovery.

## Robotic-OCT-guided laser ablation microsurgery

We utilized the robotic-OCT in combination with a laser ablation system as a microsurgical tool (see "Methods" for details) to selectively remove targeted areas of the insulating layer and expose only the relevant pins on a micro-SD card. First, to assess the accuracy and precision of robotic-OCT guided laser ablation on target areas, we conducted experiments using a total of 18 micro-SD cards of the same model (Card 8). Each card underwent ablation on ten different technological pins, with each pin considered as an individual trial. To evaluate the accuracy, we compared the center position of each ablation hole with that of the corresponding pin in the OCT image. This allowed us to determine how closely the laser ablation process aligned with the intended target. For precision assessment, we compared the center position of each ablation hole with the average center position calculated from the 18 corresponding ablation holes in each trial. This analysis provided insights into the consistency and repeatability of the laser ablation process. As shown in Fig. 6, the precision and accuracy were measured to be ±10 μm and 52 μm in the $X$ direction, and ±11 μm and 50 μm in the $Y$ direction. This achieved precision and accuracy level is sufficient for the majority of technological pin sizes.

Second, we evaluated the influence of different power levels on the laser ablation process, as shown in Fig. 7. We selected a micro-SD card sample (Card 8) contained a 6 × 6 array of technological pins on its internal PCB, with each pin having a diameter of ~300 μm. Following

OCT guidance, laser ablation was performed on each row of pins in the micro-SD card for six power levels: 2 W, 6 W, 10 W, 14 W, 18 W, and 20 W, as shown in Fig. 7a. By analyzing the en face OCT image of the card surface presented in Fig. 7b, the diameters of the ablation holes at different power levels were measured to be 75 μm, 140 μm, 170 μm, 220 μm, 230 μm, and 260 μm, respectively. In the en face OCT image at a depth of ~20 μm (Fig. 7c), the ablation holes appeared as bright spots and exhibited a progressive increase in size corresponding to the escalating power levels. Regarding ablation depth, the holes formed at the six power levels reached the maximum depths of 9 μm, 12 μm, 18 μm, 20 μm, 22 μm, and 22 μm, respectively, as shown in Fig. 7d. It was clearly demonstrated that both the size and depth of the ablation holes increased as the laser power level was raised. These results also indicated that the optimal laser power for this specific sample would be 10 W, as it would guarantee that the ablation hole size remained within the pin dimensions, and the ablation depth was close to, but did not exceed, the thickness of the insulating layer to prevent any potential damage to the PCB circuitry. This observation highlighted the importance of selecting an appropriate laser power level that strikes a balance between achieving the desired ablation results and minimizing excessive heating.

The robotic-OCT-guided laser ablation process began with OCT imaging of a small portion of the sample (Card 9) as the red rectangle shown in Fig. 8a, which produced an en face OCT image exhibiting a subgraph of the PCB trace (Fig. 8b). Next, we employed a subgraph-to-whole-graph registration method[24] based on corner detection and template matching to obtain the corresponding full-scale image of the PCB trace (Fig. 8c) from our custom-built PCB trace database of micro-SD cards. We then identified the technological pins used to access the internal flash

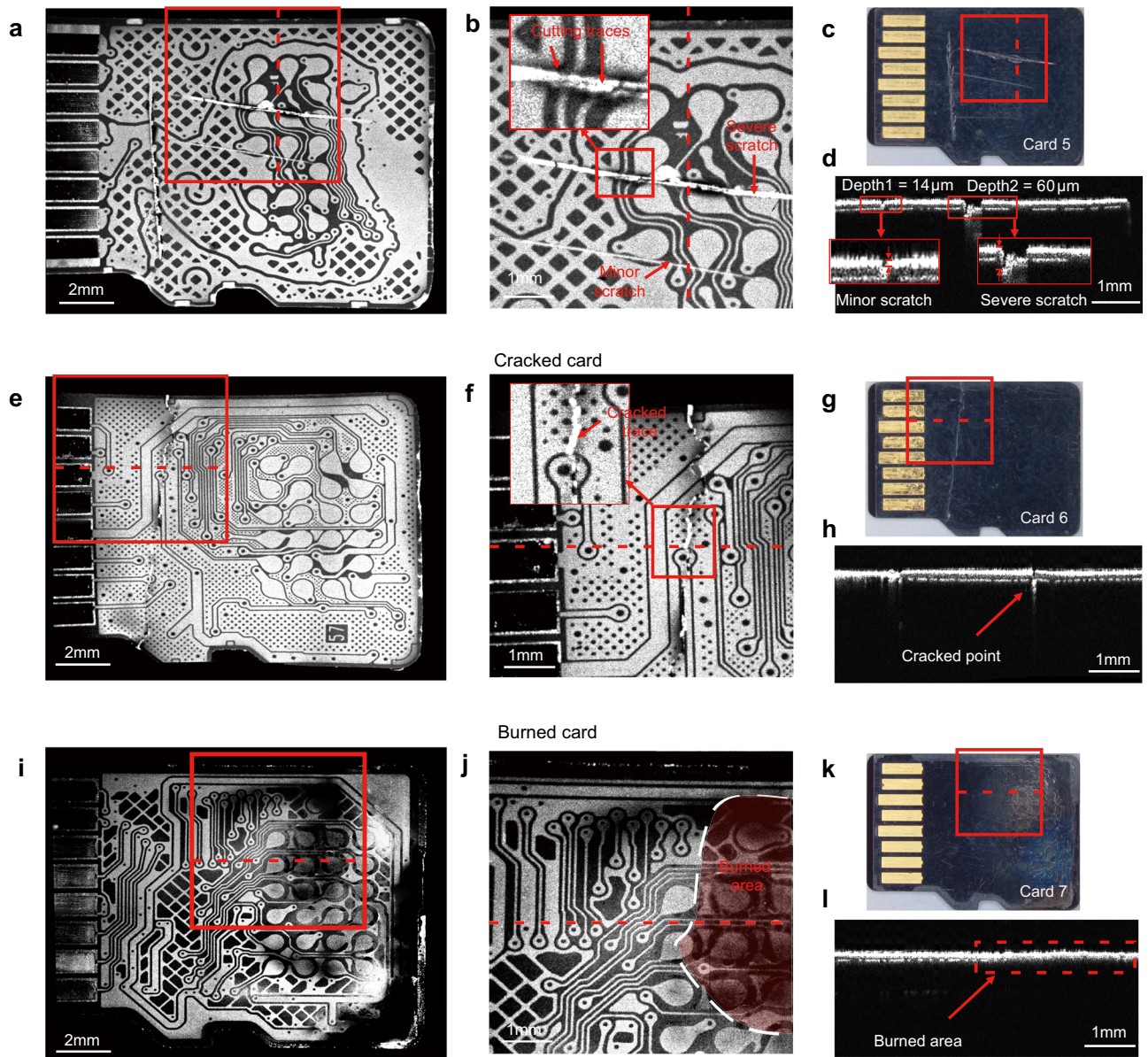

**Fig. 5 | OCT Images of scratched (Card 5), cracked (Card 6), and burned (Card 7) micro-SD Cards. a** En face OCT images of the PCB traces on the pinout side of Card 5. **b** Enlarged en face OCT images of the area indicated by the red boxes in (**a**, **c**), clearly showing minor scratch, severe scratch, and cutting traces. **c** Microscopic images of Card 5. **d** Corresponding B-scan to the red dashed lines in (**a–c**), indicating the depths of the minor and severe scratches to be approximately 14 μm and 60 μm, respectively. **e** En face OCT images of the PCB traces on the pinout side of Card 6. **f** Enlarged en face OCT images of areas indicated by the red boxes in (**e**, **g**), clearly showing the cracked trace. **g** Microscopic images of Card 6.

**h** Corresponding B-scan to the red dashed lines in (**e–g**), indicating the position of the cracked point. **i** En face OCT images of the PCB traces on the pinout side of Card 7. **j** Enlarged en face OCT images of areas indicated by the red boxes in (**i**, **k**); the red area indicates the burned area, and the white dashed line represents the boundary of the burned area. **k** Microscopic images of Card 7. **l** Corresponding B-scan to the red dashed lines in (**i–k**). The area enclosed by the red dashed line indicates the burned area, which has a different cross-sectional structure compared to the surrounding unburned area.

memory in the full-scale image and measured their diameters to be approximately 610 μm as shown in Fig. 8d. The positions of all relevant pins were recorded and transmitted to the robotic arm for precise control of its movement. The robotic arm was first actuated to move to each pin location, and then the laser burned away the insulating layer covering the corresponding pin. Finally, the exposed pins were connected to the data recovery system, enabling direct extraction of data from the internal flash memory, as shown in Fig. 8f. These results demonstrated that the robotic-OCT-guided laser ablation microsurgery could automate the removal process and precisely control the removal location to

avoid extra damage, simplifying subsequent welding processes and reducing the time and effort required to scan the entire sample.

## Robotic-OCT imaging for other types of MSDs

The robotic-OCT inspection can be extended beyond micro-SD cards to other typical MSDs such as monolithic USB flash drives and MMCs. The continuous scanning strategy was also applied to these devices as it allowed for imaging with an adjustable FOV. In this experiment, the FOV of the robotic-OCT system was adjusted to 11 mm × 30 mm and 11 mm × 15 mm for two monolithic USB flash drives (Card 9 and

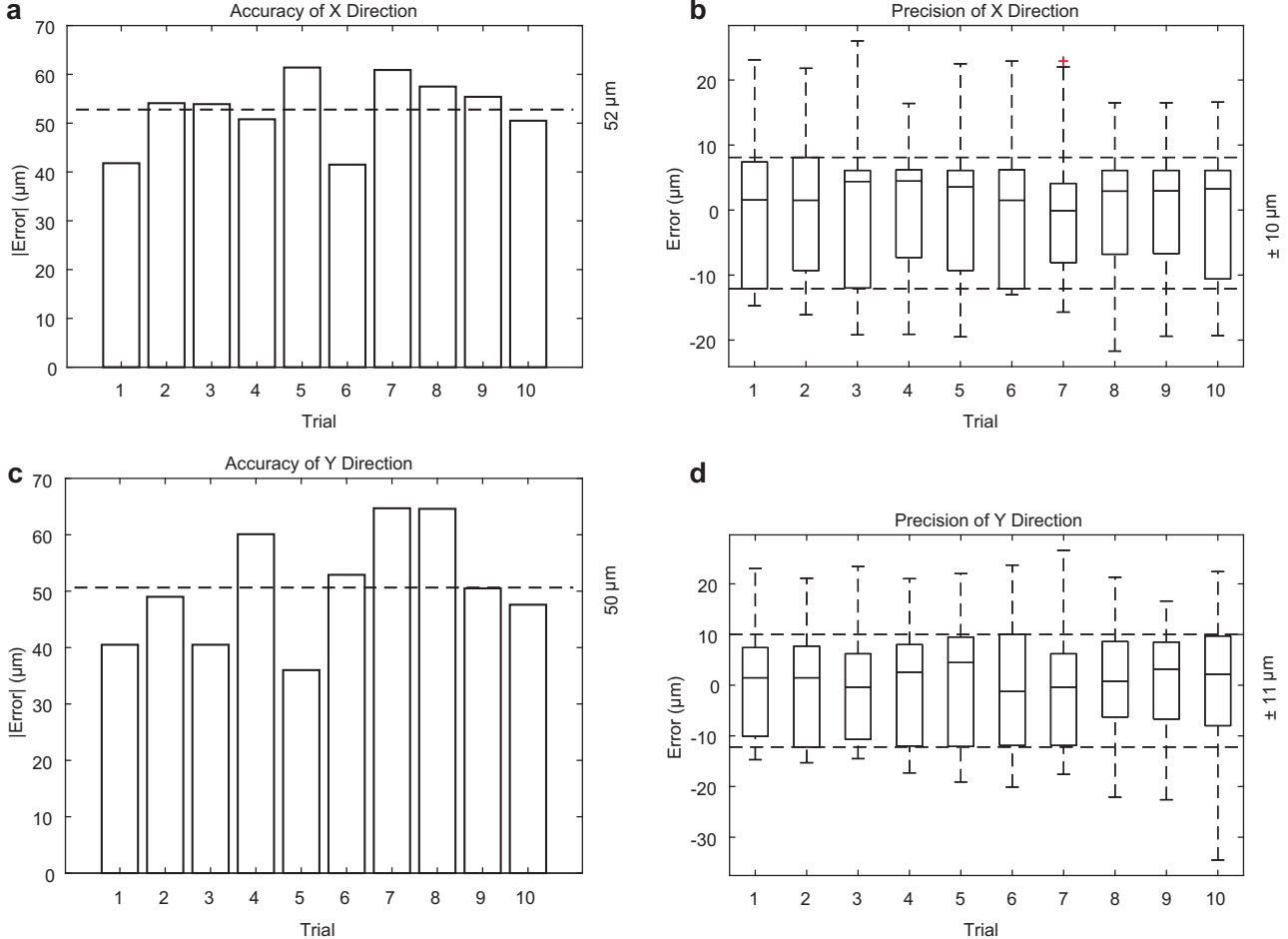

**Fig. 6 | The accuracy and precision of robotic-OCT-guided laser ablation on target areas. a** Absolute mean error for center positions of the ablation holes in the *X* (fast-axis) direction, indicating an accuracy of 52 μm (*n* = 18 samples per trial). **b** Center position distribution for the ablation holes in the *X* (fast-axis) direction, indicating a precision of ±10 μm (*n* = 18 samples per trial). **c** Absolute mean error for center positions of the ablation holes in the *Y* (slow-axis) direction, indicating an accuracy of 50 μm (*n* = 18 samples per trial). **d** Center position distribution for the ablation holes in the *Y* (slow-axis) direction, indicating a precision of ±11 μm (*n* = 18 samples per trial). For all box plots, center lines represent the median, the length of the box extends from the lower quartile to the upper quartile, whiskers are 1.5 times of interquartile range and red cross indicates outlier.

Card 10) as shown in Fig. 9a, b, which comprised two adjacent scanning regions. The FOV was adjusted to 24 mm × 42 mm for the MMC (Card 11), which composed of four adjacent scanning regions as shown in Fig. 9c. The corresponding en face images provided clear and detailed information about the PCB traces hidden beneath the insulating layer in these devices, which was valuable for the identification and analysis of the technological pins. As the FOV changed, image acquisition for these devices required 71, 37, and 200 s, respectively, which required no operator intervention. For an SD card, the essential storage component inside the SD card was still an MSD, and the corresponding robotic-OCT inspection results are shown in Supplementary Fig. 1. These results demonstrated the flexibility and versatility of the robotic-OCT system to inspect various types of MSDs.

## Discussion

This work aims at inspecting and performing microsurgery for MSDs using robotic-OCT, which addresses an important problem in the field of digital forensics and has the potential to revolutionize the data recovery procedures by replacing conventional methods that involve destructive removal of entire insulating layers or the use of X-ray inspection techniques. Our proposed method can probe the internal multilayer structure of the MSD and accurately reveal the underlying PCB traces in a non-contact, nondestructive and fast manner, which eliminates the need for manual removal of the insulating layer. Utilizing low-power (a few milliwatts) near-infrared continuous wave light, the OCT imaging technique causes no harm to the operator, the device, or the stored data. It avoids the risks of harmful radiation, potential bit errors[25,26], and laser fault injection[27], while maintaining the integrity of the device. Importantly, our robotic-OCT system acquires high-resolution data consisting of 2000 × 3000 × 2048 voxels in ~37 s. This provides a significant advantage over micro-CT, which typically takes minutes to hours for scanning similar-sized areas[28]. The detailed analysis of the insulating layer, distribution of pins and vias within an MSD, and the establishment of a comprehensive PCB trace database for various types of MSDs can be efficiently achieved through the obtained high-resolution OCT images. Furthermore, it can be utilized to identify any cracks, cuts, scratches, or burns in damaged devices, facilitating internal diagnosis of abnormal connections. This information is vital in determining the extent of damage, evaluating the circuit's integrity, and assessing the repairability of the device. Consequently, the nondestructive high-speed robotic-OCT imaging can maximally preserve the sample's integrity, greatly reduce manual labor, and significantly increase the chances of successful data recovery from the MSD.

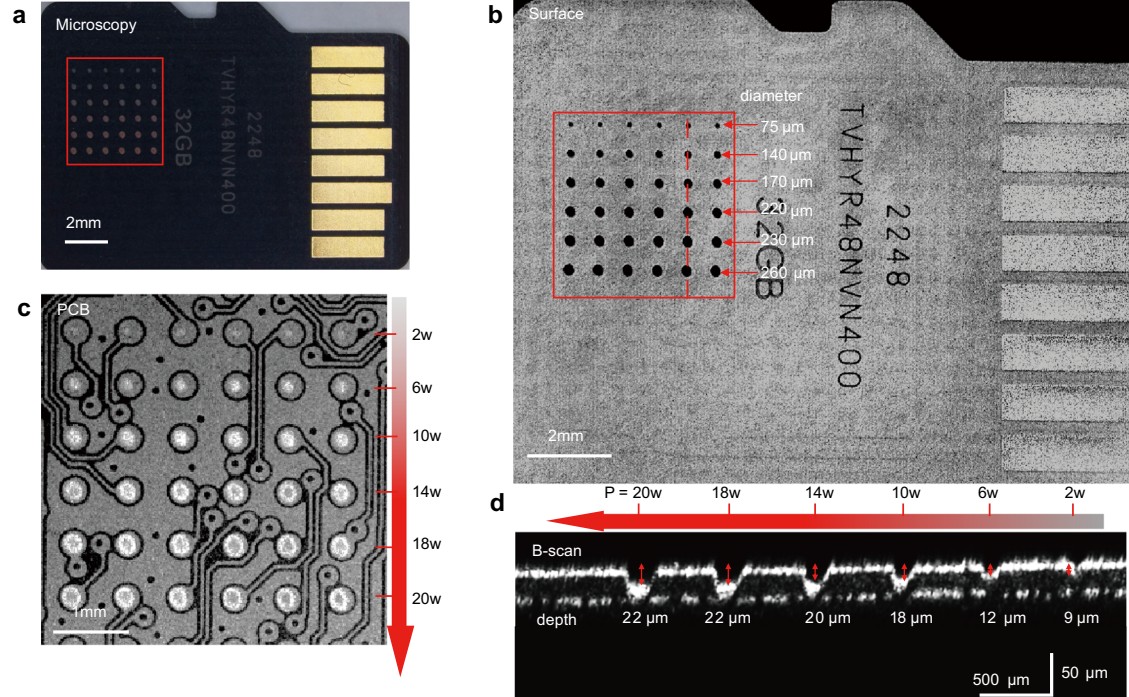

**Fig. 7 | Evaluation of different power levels on the laser ablation process.**
**a** Microscopic image showing a micro-SD card ablated with six different power
levels. **b** En face OCT image of the ablated card surface; the diameters of the
ablation holes for six power levels (2 W, 6 W, 10 W, 14 W, 18 W, and 20 W) are
measured to be 75 μm, 140 μm, 170 μm, 220 μm, 230 μm, and 260 μm, respectively.
**c** En face OCT image of the ablated card at a depth of -20 μm (pinout side of the

PCB), demonstrating the precise positioning and different sizes of the ablation
holes at six power levels (2 W, 6 W, 10 W, 14 W, 18 W, and 20 W). **d** B-scan at the red
dashed lines in (**b**), indicating the maximum depths of the ablation holes at six
power levels (2 W, 6 W, 10 W, 14 W, 18 W, and 20 W) to be 9 μm, 12 μm, 18 μm,
20 μm, 22 μm, and 22 μm, respectively.

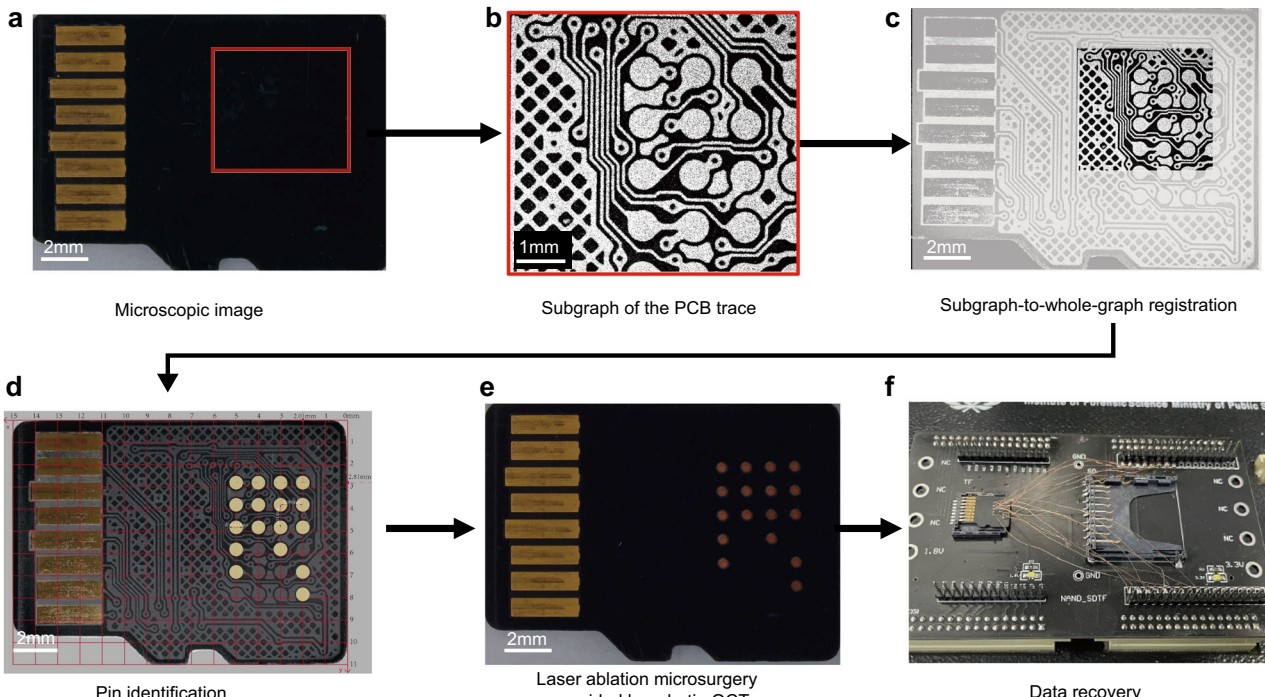

**Fig. 8 | Robotic-OCT-guided laser ablation microsurgery of a micro-SD card.**
**a** Microscopic image of Card 8. **b** En face OCT image of the area indicated by the red
box in (**a**). **c** Result of subgraph-to-whole-graph registration. **d** Identification and
measurement of the technological pins; a grid line scale is added. **e** Micro-SD card

after laser ablation microsurgery with only the relevant pins exposed. **f** Accessing
data directly from the internal flash memory by connecting the exposed pins to the
data recovery system.

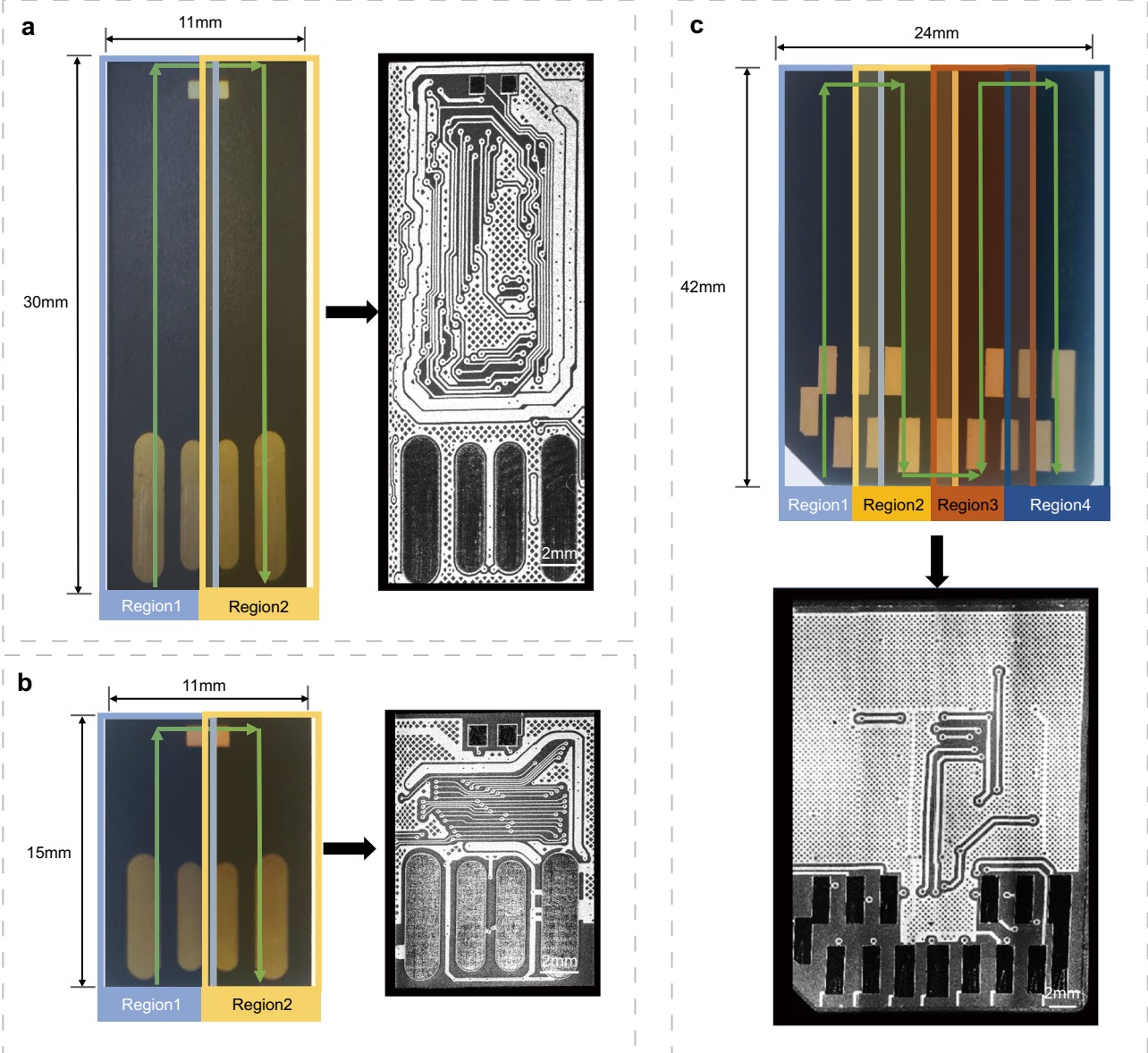

**Fig. 9 | Robotic-OCT imaging of monolithic USB flash drives and MMC.**
**a** Monolithic USB flash drive (Card 9) and the corresponding en face OCT image with a FOV of 11 mm × 30 mm; the green arrow line indicates the scanning trajectory of the robotic arm and the two adjacent continuous scanning regions, represented by blue (Region 1) and yellow (Region 2). **b** Monolithic USB flash drive (Card 10) and the corresponding en face OCT image with a FOV of 11 mm × 15 mm; the green arrow line indicates the scanning trajectory of the robotic arm and the two adjacent continuous scanning regions, represented by blue (Region 1) and yellow (Region 2). **c** MMC (Card 11) and the corresponding en face OCT image with a FOV of 24 mm × 42 mm; the green arrow line indicates the scanning trajectory of the robotic arm and the four adjacent continuous scanning regions, represented by light blue (Region 1), yellow (Region 2), red (Region 3), and dark blue (Region 4).

One limitation of the robotic-OCT inspection method is the difficulty in obtaining PCB traces on the dice side due to the high reflectivity of the copper layer on the pinout side, which obstructs visibility. Although the PCB traces on the pinout side contain the majority of the relevant information for data recovery, information from the dice side remains valuable for a comprehensive analysis. A potential solution to address this limitation is to perform imaging from the dice side, rather than from the pinout side in our experiment. However, the thick plastic substrate on the dice side hinders light penetration, especially when using a near-infrared OCT system with limited penetration depth. To overcome this challenge, mid-infrared OCT[29, 30] can be employed, which reduces scattering effects and improves penetration through the thicker plastic substrate. This would enable better visualization of PCB traces on the dice side. However, it introduces another challenge that the electronic components and

copper layers on the dice side of the PCB may impede mid-infrared light penetration, affecting the visibility of PCB traces on the pinout side. For effective visualization of PCB traces on both sides, it is favorable to use mid-infrared OCT separately for each side. Meanwhile, it is crucial to consider the limitations associated with mid-infrared OCT, including potential degradation of spatial resolution, lower signal-to-noise ratio (SNR), and increased complexity and costs. Besides, a basic OCT imaging system typically costs tens of thousands of dollars, and the addition of a robotic arm and other components necessary for automated scanning and imaging can increase the size and cost. However, the size and cost of an OCT system can be decreased using integrated optics, a suitable fabrication technology, and an optimum optical design.

Moreover, while it is true that galvo-based OCT scanners have been previously attached to robotic arms or translation stages to

extend the field-of-view (FOV)[22, 31–33], they typically employ a stop-and-stare scanning approach, which is time-consuming and inefficient due to the frequent restarts of the robotic arm or translation stage, as well as the subsequent stitching of data from each scanned block. Unlike the conventional stop-and-stare scanning approach, our study introduces a continuous scanning strategy specifically designed for the robotic-OCT system, where the robotic arm moves continuously over the object and captures data. By eliminating the need for start-stop motions between individual scans, our continuous scanning strategy significantly reduces scanning time and enables large areas to be scanned more efficiently than with conventional methods. For instance, the stop-and-stare approach typically requires ~11 min to obtain $1024 \times 256 \times 18$ A-lines, including pose optimization and manual scanning point selection operation[22]. In contrast, our continuous scanning strategy is capable of acquiring the same number of A-lines in ~30 s, resulting in a 14-fold increase in scanning speed at the same A-line rate. Moreover, the continuous movement of the robotic arm ensures uniform and seamless image acquisition, significantly eliminating the brightness variations and motion artifacts commonly found in traditional stop-and-stare methods, and enhancing the overall image quality. Furthermore, the strategy can be extended to accommodate larger or irregularly shaped objects by precisely controlling the robotic arm to reposition the scanner at various positions and angles. This adaptability enables customized scanning trajectories based on the unique shape and size of the sample, making the imaging process versatile across a wide range of applications, such as diagnosing and treating organs or tissues, as well as documenting and analyzing artifacts. Nevertheless, one significant limitation of the continuous scanning strategy is the management of the large amount of data generated during a single scan, especially when imaging larger objects. This can pose challenges in terms of data storage, processing, and analysis. Efficient data processing techniques and robust storage solutions are vital to effectively handle the increased data throughput. Another limitation is the accurate capture of geometric shape and surface details of objects to plan scanning trajectories, particularly for irregularly shaped ones. This limitation can be addressed by incorporating additional imaging technologies, such as a 3D camera, into the robotic-OCT system to provide accurate spatial information.

Furthermore, inspired by image-guided surgical interventions in the medical field, we have developed a robotic-OCT-guided laser ablation technique—Chip Surgery Robot. Robotic assistance ensures precise and controlled movements during the laser ablation process, while OCT provides high-resolution sub-surface imaging and guidance, along with quantitative information to evaluate the ablation process. This advanced technique enables precise microsurgery for MSDs by accurately and automatically removing unwanted layers or structures while minimizing sample damage. By doing so, it selectively exposes the relevant pins necessary for data recovery from the flash memory, eliminating the need to fully expose the entire PCB as done in conventional methods. As a result, this approach offers several advantages, including simplifying the subsequent welding process, and allowing operators to focus on specific areas without concerns about the rest of the sample or the risk of a short circuit. However, it is important to note that the precision and accuracy of the laser ablation process determine the minimum pin size that can be accurately and reliably exposed. Our current system achieves accurate and reliable laser ablation for pin sizes above 100 μm. To further enhance the accuracy and precision, we can employ higher-resolution OCT systems and higher-precision robotic arms that can provide more detailed and precise guidance during the procedure, enabling better visualization of the target area and facilitating improved alignment of the laser beam with the pins. Meanwhile, it is essential to choose a laser power level that aligns with the specific diameter of the pin and the thickness of the insulating layer, which ensures the creation of suitably sized

holes to facilitate subsequent processes such as welding and wiring during data recovery, while minimizing any potential damage to the internal circuitry. In addition, human intervention is still required during the welding and wiring process to perform data recovery. To automate the entire process, an automated robotic welding system can be combined with the current system to address this issue in our follow-up work.

While OCT cannot trace the bonding wires of the flash memory die, it can effectively capture the image of the PCB layout on the pinout side of the MSD. This image can then be matched with entries in our current PCB layout database. If a matching layout is found, it signifies the presence of that specific MSD model in the database, enabling us to directly identify the relevant technological pins. However, if the MSD model is not present in the database, the pin assignments can still be determined by selectively exposing all the technological pins using robotic-OCT-guided laser ablation microsurgery and connecting each exposed pin to a logic analyzer for function analysis. This procedure eliminates the need for completely removing the insulating layer and ensures minimal damage to the MSD sample.

Finally, although our work focuses on a specific application (data recovery from MSDs), it has the potential to be extended to various applications across different fields. For example, the proposed robotic-OCT-guided laser ablation technique can be utilized for precise and real-time identification of tumor margins during surgical procedures. Surgeons can accurately target and remove cancerous tissue while minimizing damage to healthy surrounding tissue. In the field of microelectronics, our technique's ability to precisely remove unwanted layers can be valuable for the fabrication of complex integrated circuits, where selective material removal is crucial for optimizing circuit performance. It also opens up new possibilities for broader applications in digital forensics, failure analysis, materials testing, and quality control, bringing advancements in precision and efficiency.

In summary, the robotic-OCT represents a significant advance for recovering data from MSDs, overcoming the drawbacks of the conventional methods by providing nondestructive, high-resolution, high-speed and flexible tomographic imaging. It can non-destructively reveal the internal PCB traces, locate and identify the technological pins and vias, and diagnose the damaged devices. Targeted areas of the insulating layer can also be removed by robotic-OCT-guided laser ablation microsurgery automatically and precisely, exposing only the relevant pins to facilitate subsequent welding and data extraction. Although this work has mainly been carried out in the context of micro-SD cards, the same approach has been successfully applied to other major types of MSDs. Thus, we foresee a promising role for robotic-OCT as a valuable and reliable technique in various fields such as digital forensics, failure analysis, materials testing, and quality control.

## Methods

### OCT engine

We built a customized spectral-domain OCT engine as shown in Supplementary Fig. 2a with ~4 μm axial resolution in air, ~7 μm lateral resolution and 2.27-mm imaging depth. The engine employed a super luminescent diode (SLD; IPSDW0825, InPhenix) as the light source centered at 850 nm with 105 nm −3dB spectral bandwidth. A custom-built k-linear spectrometer with a F2 prism (PS852, Thorlabs), a 1200 lines/mm diffraction grating (WP-1200/840-25.4, Wasatch Photonics), a fiber-coupled collimator (RC08APC-P01, Thorlabs) and an achromatic lens (#49-381, Edmund) was used to record the linear-in-wavenumber interferogram and eliminate the need for interpolation. The custom-built objective lens of the OCT probe has a diameter of 25.4 mm, a focal length of 25 mm and a working distance of 20 mm. The system's signal-to-noise ratio (SNR) was measured to be 110 dB, with 4.5 mW optical power at the sample and 8.3-μs integration time for the 2048-pixels 12-bit line-scan CMOS camera (Octoplus, Teledyne

e2v, UK), corresponding to 120 kHz A-line rate. We obtained B-scan images by fast-axis lateral scanning, with a duty cycle of 95%, containing 1000 A-lines, resulting in a B-scan frame rate of ~114 Hz.

## Robotic-OCT scanner

The robotic-OCT scanner was constructed by attaching an OCT probe to a six degrees of freedom (DOF) robotic arm (Dobot, China) using a custom-designed adapter. The OCT probe consisted of a fiber collimator, a galvo scanner and an objective lens. The robotic arm had a payload of 3 kg, a repeatable positioning accuracy of ±0.02 mm, and an arm span of 620 mm. We employed a two-step hand-eye calibration method to align the robotic arm and OCT probe with the imaging target and keep the sample within the OCT system's depth of focus[20]. First, we imaged a needle with a tip size less than 10 μm to locate the OCT probe's focal point via orthogonal B-scan scanning. This allowed us to determine the 3D position of the focal point and its 3D position relative to the robotic arm's flange plane. Second, we acquired OCT images of shell calibration targets at different views and registered them using the iterative closest point (ICP) algorithm to get the transformation matrix from the robotic end-effector to the OCT frame. After the calibration, the moving direction of the robotic arm was set perpendicular to the fast-axis direction provided by the galvo to avoid any potential deformation or distortion in the acquired images.

The scanning speed of the robotic arm was set at ~1 mm/s to maintain consistent lateral resolution in both orthogonal scanning directions. The total data acquisition time to scan a FOV of 15.5 × 11.3 mm, corresponding to 3000 × 1900 pixels, was ~37 s. This included two continuous scans and the turnaround time (the time for the robotic arm to reverse and scan in the opposite direction). During the scanning, we tracked the positions of the robotic arm to locate each B-scan and determine the B-scan numbers to arrange the acquired B-scans in the right order. The 3D volumetric image could be automatically reconstructed from the acquired 2D B-scans, with the B-scans stacked in their correct positions to form a volume.

## Laser ablation microsurgery system with robotic-OCT guidance

A high-energy nanosecond fiber laser (10 ns, 20 W, 532 nm) was integrated to the robotic-OCT system to perform laser ablation microsurgery and remove the targeted insulating layer, as shown in Supplementary Fig. 2a. The laser shared the same optical path as the robotic-OCT scanner through a short-pass dichroic mirror (Thorlabs, USA), as shown in Supplementary Fig. 2b. The dichroic mirror had a cutoff wavelength of 650 nm and separated the OCT imaging light from the laser light, enabling simultaneous alignment of the laser beam and OCT imaging, as well as precise positioning and targeting of ablation. The laser power was set to be 10 W which could be varied depending on the thickness of the insulating layer. The robotic-OCT imaging provided accurate feedback on the location of the technological pins, allowing for selective removal of the insulating layer over the pins while avoiding damage to other parts of the sample.

## Data and image processing

The raw spectral OCT data were processed according to standard spectral-domain OCT techniques including DC subtraction, dispersion compensation and inverse discrete Fourier transform to generate the A-scans. The B-scans were then generated by stacking the A-scans and displaying them in log-scale to enhance the contrast and dynamic range. Additionally, the B-scans were rescaled to adjust the brightness and contrast. The identical white and black thresholds were applied to all B-scans.

Due to the field curvature resulted from the scanning of galvanometer[34], we developed an algorithm to compensate the

curvature aberration, which mainly consists two steps: surface detection and curve plane fitting. Surface detection was implemented according to Eq. (1),

$$Z(x_n, y_m) = z' \text{ when}$$
$$g_{(10 \times 10)} \bigotimes \{ I((x_n, y_m, z') > \text{Max}(x_n, y_m) \times \text{threshold}) \} \quad (1)$$

where $Z$ indicates the derived depth of surface points in each A-scan, $n$ and $m$ represent the frame number in the $x$ and $y$ directions, $g$ represents a 2D gaussian filter with a 10 × 10 convolutional kernel to reduce the impact of noise and smooth the surface boundary. Then a third-degree polynomial curve plane function was used to fit the detected surface. Finally, the curvature could be compensated and the en face OCT image displaying the PCB traces could be obtained via a single slice. As two adjacent continuous scanning regions were required to image a micro-SD card, we stitched the two corresponding en face images based on the recorded positions of the robotic arm and the corresponding B-scan numbers, and applied a multi-band blending algorithm to remove the suture seams and illumination discrepancies. We used the same method to create the OCT images for other MSDs. The robotic arm and laser ablation, as well as the generation of A-scans, B-scans, and 3D volumetric images, were controlled using custom software written in C++.

## Insulating layer characterization

To calculate the thickness of the insulating layer from OCT images, we first measured the distance between two adjacent boundaries representing the OPL of the insulating layer. The OPL can be estimated from the axial resolution of the OCT system and the number of pixels between the boundaries. As the OPL is the product of the refractive index of the sample and its geometric thickness, we obtained the thickness of the insulating layer by dividing the OPL by the assumed refractive index of ~1.5, which is typical for polymer films. To increase the accuracy of the calculated thickness, we manually selected 30 B-scans from the 3D volume data of the sample and obtained 30 measurements for each B-scan. The thickness was calculated for each measurement and averaged across all measurements to obtain a more accurate value. Meanwhile, the STD was estimated through all measurements to quantify the degree of variation in the thickness of the insulating layer. The number of B-scans and measurements can be varied depending on the desired level of accuracy and the size of the sample.

## Safety precautions

First, all personnel operating the laser were required to wear safety glasses specifically designed for the laser's wavelength. This protected their eyes from potential injury caused by laser beams. Additionally, laser power settings were carefully adjusted within safe operating limits to achieve the desired ablation outcome while minimizing any potential risk to the sample or operator. Regular monitoring of the laser power output was conducted to maintain consistent and safe settings. Warning signs were prominently displayed, and unnecessary reflective surfaces were removed from the working area to prevent accidental reflections that could cause harm. Physical barriers and marked safety zones were established around the robotic arm to prevent accidental contact and enable immediate halting of the arm's operation in emergencies. Moreover, the robotic arm's posture was restricted to ensure that the laser beam was directed away from personnel and critical areas. Finally, comprehensive safety training was provided to all individuals involved, covering laser safety protocols, emergency procedures, and safe operation of the equipment.

## Sample preparation

Twelve types of micro-SD cards from multiple sources were collected as representative samples in our experiments (see Supplementary Table 1). Card 1 was a brand-new BanQ 64GB micro-SD card obtained directly from the manufacturer. Card 2 and Card 3 were two used micro-SD cards obtained from the second-hand markets, labeled as Kingston 16GB micro SDHC class 10 and Kingston 16GB micro SDHC class 4, respectively. Card 4 was seized from a real case without any brand information. Card 5, Card 6, and Card 7 were three damaged cards to simulate common real situations—scratched, cracked, and burned, labeled as SanDisk 16GB micro SDHC class 4, Kingston 2GB micro-SD and TOSHIBA 16GB micro SDHC class 4, respectively. Among these eight samples, Card 2 presented a relatively glossy surface on the pinout side, and the PCB trace could be vaguely observed through microscopy. In contrast, the other samples presented matte surfaces, and the PCB traces could hardly be seen by microscopy. Card 8 was a brand-new Kingston 32GB micro-SD card for the evaluation of the laser ablation process. Card 9 was obtained from the second-hand markets and used to demonstrate robotic-OCT-guided laser microsurgery of micro-SD card to erase the insulating layer. Card 10, Card 11, and Card 12 were monolithic USB flash drives and MMC obtained directly from the manufacturers. A diverse set of over 80 micro-SD cards, including different brands and models collected from multiple sources (see Supplementary Fig. 3) were collected to assess the scattering effects of the insulating layers.

## Data availability

All data supporting the findings of this study are available within this paper and its Supplementary information file. The raw OCT data, due to their large file size, are available from the corresponding author upon request.

## Code availability

The robotic-OCT software, including the data acquisition, post-processing, and analysis code, is available from the corresponding author upon request as the terms of the license established prior to this study with all collaborators do not allow uploading the code in a public repository.

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

## Acknowledgements

This research was sponsored by the Beijing Nova Program of Science and Technology 20220484179 (N.Z.), Z191100001119039 (N.Z.), the National Key Research and Development Program 2021YFF0602105 (N.Z.), the Central Public-interest Scientific Institution Basal Research Fund 2023JB015 (L.Z.), the National Natural Science Foundation of China 61975091 (P.X.), 61905015 (P.X.).

## Author contributions

B.H., Y.X.Z., and L.Z.: designed the robotic-OCT system, developed the custom software, system calibration, wrote the original article; Z.W.S. and Y.R.K.: interpretation of 2D en face OCT images, pinout analysis, sample acquisition, data recovery, and laser ablation integration; X.Y.H: theoretical development of the OCT image processing algorithms; L.W., Z.H.L., W.H., Z.G.L., G.D.X., and F.H.: analysis of OCT data and evaluation of laser ablation process; C.M.W.: constructed the probe; P.X.: manuscript reviewing and funding acquisition; N.Z.: conceptualized the robotic-OCT system and designed all the experiments, manuscript reviewing and editing, funding acquisition, and supervised the project.

## Competing interests

The authors declare no competing interests.
