## [Peer Review File · Nature Communications]

Robotic-OCT guided inspection and microsurgery of monolithic storage devicesReviewer #1 (Remarks to the Author):

In this study, the authors developed an OCT system for the inspection and microsurgery of monolithic storage devices. To examine such devices, conventional OCT does not provide a sufficient field of view. To address the unmet need, the authors attached the OCT scanner to a robotics arm and assemble macroscopic images. This manuscript presented non-invasive inspection of MSDs with high-quality OCT images. The work presented by this manuscript targeted a very specific application and the authors provided strong data to demonstrate the usefulness of the imaging platform. The methodology is sound, and the conclusions are supported by strong experimental data. However, this study has limited novelty. The major weakness of this study is that it does not bring new knowledge or new methods to answer scientific questions. At most, it addresses an engineering problem within a very narrow scope. OCT technology is a well-established imaging modality and galvo-based OCT scanners have been attached to other scanning devices to extend its FOV (Huang et al, Biomed. Opt. Express 12, 4596-4609 (2021); Sprenger et al., 2021 IEEE 18th International Symposium on Biomedical Imaging, 1137-1140; Finke et al, The International Journal of Medical Robotics and Computer Assisted Surgery, 8(3), pp.327-336, 2012; Callewaert et al Opt. Express 28, 26239-26256 (2020)). Moreover, some storage devices have a large protective enclosure with a thick layer of plastic. Does the image quality change if a larger SD card has a thicker protective layer?

Reviewer #2 (Remarks to the Author):

General Comments:

The manuscript titled "Robotic-OCT Guided Inspection and Microsurgery of Monolithic Storage Devices" presents a novel approach for non-destructive inspection and microsurgery of monolithic storage devices (MSDs) using a robotic-arm-assisted optical coherence tomography (robotic-OCT) system. The authors describe the capabilities of the system in rapid imaging, defect detection, pin identification, and laser ablation for targeted area removal. The manuscript highlights potential applications in digital forensics, failure analysis, materials testing, and quality control. Overall, the work is of significant novelty and addresses the need for non-destructive techniques in data recovery, and merits the publication. Nevertheless, there are a few points that require further clarification and improvement.

Specific Comments:

ABSTRACT

The abstract outlines the key features of the robotic-OCT system, including its ability to facilitate high-resolution imaging, PCB visualization, pin identification, defect detection, and selective removal of targeted areas using laser ablation. The abstract also highlights the diverse potential applications of the robotic-OCT technology in digital forensics, failure analysis, materials testing, and quality control. This is an interesting and very practical application topic. However, there are a few suggestions to enhance its clarity and completeness:

1. Quantify the benefits: Provide quantitative information on the benefits offered by the robotic-OCT system. For example, mention the achievable imaging resolution, the reduction in inspection time compared to traditional methods, and any improvements in data recovery efficiency observed in the study. Quantitative data will add credibility and demonstrate the practical advantages of the system.

2. Expand on the laser ablation technique: Discuss the precision and accuracy of the laser ablation process, potential challenges, and any considerations for ensuring minimal damage to the device during microsurgery.

Introduction

The Introduction section provides a comprehensive overview of the challenges and limitations associated with current methods of data recovery from monolithic storage devices (MSDs). It introduces optical coherence tomography (OCT) as a potential solution and highlights its advantages over X-ray radiography. The integration of a robotic arm and the use of continuous scanning are also mentioned as important components of the proposed approach. Overall, the Introduction sets the stage for the study and establishes the motivation for developing the robotic-

OCT system. It is a high-quality introduction that deserves praise. However, here are a few suggestions to further improve the Introduction:

1. Address potential concerns: Given that OCT is traditionally used in the biomedical field, it may be beneficial to briefly address any potential concerns or challenges in applying OCT to non-biomedical applications such as MSD inspection. This could include factors like sample size, surface reflectivity, or other limitations that might affect the imaging quality or feasibility of the approach.

2. Clarify the novelty: Although the introduction mentions that this study is a significant improvement over traditional methods, explicitly emphasizing the novelty of this study can accurately highlight the research value more and contribution of the article and enhance the reader's awareness of the novelty and importance of the article. For example, the need and advantages of combining robotics with optical coherence tomography are emphasized.

Methods

The Methods section in this article is well-organized, detailed, and provides a comprehensive overview of the experimental procedures and techniques used to develop the custom-built robotic-OCT system. However, here are a few suggestions to further improve the Introduction:

1. It would be helpful to include more information about the specific components and manufacturers of the OCT engine, such as the brand and model of the super luminescent diode (SLD) and the k-linear spectrometer, OCT probe objective lens size, focal length, working distance. This additional detail will provide a clearer understanding of the setup and equipment used in the study.

2. It would be valuable to mention any safety measures or precautions taken during the laser ablation microsurgery procedure. This could include details about laser power settings, safety protocols, and any measures implemented to prevent damage to the sample or ensure operator safety.

3. Some graphs, such as Fig. 6 lack scale bars.

Result

The results section clearly explains the details of the customized robotic-OCT system and the experimental procedure. Here are a few suggestions to further improve the Introduction:

1. Clarify the specific benefits and advantages of the continuous scanning strategy. How does this strategy improve the imaging process compared to other scanning methods? Highlighting these advantages and discussing any limitations or trade-offs would provide a more comprehensive evaluation of the approach.

2. Address the feasibility of extending the continuous scanning strategy to larger or irregularly shaped objects. Are there any practical limitations or challenges that need to be considered? Discussing the scalability and adaptability of the approach would contribute to the broader applicability of the robotic-OCT system.

3. Additionally, it would be valuable to discuss the potential limitations or challenges encountered during the microsurgery procedure, such as the precision of pin exposure or any thermal effects caused by laser ablation.

Discussion

The authors' discussion highlights the novelty and potential impact of their work on the field of data recovery from microsurgery for MSDs. They provide a clear contrast between the traditional methods of data recovery, which are time-consuming, require expertise, and can cause damage to the samples, and the proposed robotic-OCT system, which enables automated, non-destructive, and fast imaging. The authors rightly point out that their work has the potential to revolutionize data recovery procedures for MSDs.

Reviewer #3 (Remarks to the Author):

The authors do a good job introducing the optical coherence tomography as a safe and fast option for inspecting the internal PCB layers of the monolith flash storage devices which can substitute the traditional X-ray inspection. The authors also automated the whole process which can help

make forensic data extraction more efficient than before.

This research can be a good reference for digital forensics labs to introduce OCT instead of X-ray for reverse-engineering the target device PCBs. Also, this work can contribute to make the data acquisition process quicker than before in digital forensics.

However, one thing I feel missing is the detailed evaluation of the OCT inspection. The authors do mention the limitation saying that the 2nd layer of the PCB cannot be observed due to the thickness of the epoxy. But up to how many micro meters can the OCT penetrate to observe further layers? If there are more than 2 layers of PCBs, then how many layers can be inspected? And does the material of the epoxy affect the efficiency of OCT inspection? Those details should be discussed in the article. The authors suggest using the mid-infrared OCT, but no further detail is discussed. Please make it clear what becomes better and what limitation the operators would face by changing the wavelength of the infrared.

Overall, the article lacks detailed explanation of each steps and investigations. For example, in section 2.5, the authors inspect variable damaged SD cards, but its results are pretty vague. For example, for the scratched card, how deep is the scratch based on the OCT inspection? Is it reaching the flash die and the data is not recoverable? Other sections are also a bit too much summarized. Please try to be more descriptive.

The work itself is nice and helpful for forensic investigation. Since applying OCT in device inspection is the main part of this work (in addition to automating the laser ablation process), the readers would expect more evaluation details.

Also please reconsider the structure of the article. The whole time I was missing the background information (the one mentioned in section 4) and it was hard to follow all the sections.

Point-by-point response to the Reviewers' comments

We thank the three Reviewers for their valuable comments and suggestions. We have revised the manuscript and provided new results to address all the concerns raised.

REVIEWER COMMENTS

Reviewer #1 (Remarks to the Author):

In this study, the authors developed an OCT system for the inspection and microsurgery of monolithic storage devices. To examine such devices, conventional OCT does not provide a sufficient field of view. To address the unmet need, the authors attached the OCT scanner to a robotics arm and assemble macroscopic images. This manuscript presented non-invasive inspection of MSDs with high-quality OCT images. The work presented by this manuscript targeted a very specific application and the authors provided strong data to demonstrate the usefulness of the imaging platform.

Response: We appreciate the reviewer's acknowledgment of the significance of our work in addressing the unmet need in this field and the strong data presented in our manuscript. Please find below the answers to the specific questions.

- 1) *The methodology is sound, and the conclusions are supported by strong experimental data. However, this study has limited novelty. The major weakness of this study is that it does not bring new knowledge or new methods to answer scientific questions. At most, it addresses an engineering problem within a very narrow scope. OCT technology is a well-established imaging modality and galvo-based OCT scanners have been attached to other scanning devices to extend its FOV (Huang et al, Biomed. Opt. Express 12, 4596-4609 (2021); Sprenger et al., 2021 IEEE 18th International Symposium on Biomedical Imaging, 1137-1140; Finke et al, The International Journal of Medical Robotics and Computer Assisted Surgery, 8(3), pp.327-336, 2012; Callewaert et al Opt. Express 28, 26239-26256 (2020)).*

Response:

We sincerely appreciate the reviewer's comments and would like to address the concerns regarding the novelty of our work. We would like to highlight the novel contributions of our study as follows.

Novel contribution 1: While it is true that galvo-based OCT scanners have been previously attached to robotic arms or translation stages to extend the field of view (FOV) [22][31-33], they typically employ a stop-and-stare scanning approach, which is time-consuming and inefficient due to the frequent restarts of the robotic arm or translation stage, as well as the subsequent stitching of data from each scanned block. Unlike the conventional stop-and-stare scanning approach, our study introduces a novel continuous scanning strategy specifically designed for the robotic-OCT system, where the robotic arm moves continuously over the object and captures data. By eliminating the need for start-stop motions between individual scans, our continuous scanning strategy significantly reduces

scanning time and enables large areas to be scanned more efficiently than with conventional methods. For instance, the stop-and-stare approach typically requires ~11 minutes to obtain 1024×256×18 A-lines, including pose optimization and manual scanning point selection operation [22]. In contrast, our continuous scanning strategy is capable of acquiring the same number of A-lines in ~30 seconds, resulting in a remarkable 14-fold increase in scanning speed at the same A-line rate. Moreover, the continuous movement of the robotic arm ensures uniform and seamless image acquisition, significantly eliminating the brightness variations and motion artifacts commonly found in traditional stop-and-stare methods, and enhancing the overall image quality. Furthermore, the strategy can be extended to accommodate larger or irregularly shaped objects by precisely controlling the robotic arm to reposition the scanner at various positions and angles. This adaptability enables customized scanning trajectories based on the unique shape and size of the sample, making the imaging process versatile across a wide range of applications, such as diagnosing and treating organs or tissues, as well as documenting and analyzing artifacts. **In the revised manuscript, we have emphasized this novelty of our research in the Discussion section (line 421 - 441).**

Novel contribution 2: This is the first work aimed at inspecting and performing microsurgery for MSDs using robotic-OCT, which addresses an important problem in the field of digital forensics and has the potential to revolutionize the data recovery procedures by replacing conventional methods that involve destructive removal of entire insulating layers or the use of X-ray inspection techniques. Our proposed method can probe the internal multilayer structure of the MSD and accurately reveal the underlying PCB traces in a non-contact, non-destructive and fast manner, which eliminates the need for manual removal of the insulating layer. Utilizing low-power (a few milliwatts) near-infrared continuous wave light, the OCT imaging technique causes no harm to the operator, the device, or the stored data. It avoids the risks of harmful radiation, potential bit errors [25][26], and laser fault injection [27], while maintaining the integrity of the device. Importantly, our robotic-OCT system acquires high-resolution data consisting of 2000 × 3000 × 2048 voxels in ~ 37 seconds. This provides a significant advantage over micro-CT, which typically takes minutes to hours for scanning similar-sized areas [28]. The detailed analysis of the insulating layer, distribution of pins and vias within an MSD, and the establishment of a comprehensive PCB trace database for various types of MSDs can be efficiently achieved through the obtained high-resolution OCT images. Furthermore, it can be utilized to identify any cracks, cuts, scratches, or burns in damaged devices, facilitating internal diagnosis of abnormal connections. This information is vital in determining the extent of damage, evaluating the circuit's integrity, and assessing the reparability of the device. Consequently, the non-destructive high-speed robotic-OCT imaging can maximally preserve the sample's integrity, greatly reduce manual labor, and significantly increase the chances of successful data recovery from the MSD. **In the revised manuscript, we have emphasized this novelty of our research in the Discussion section (line 380 – 400).**

Novel contribution 3: Inspired by image-guided surgical interventions in the medical field, we have developed a robotic-OCT-guided laser ablation technique called the "Chip Surgery

Robot". Robotic assistance ensures precise and controlled movements during the laser ablation process, while OCT provides high-resolution sub-surface imaging and guidance, along with quantitative information to evaluate the ablation process. This advanced technique enables precise microsurgery for MSDs by accurately and automatically removing unwanted layers or structures while minimizing sample damage. By doing so, it selectively exposes the relevant pins necessary for data recovery from the flash memory, eliminating the need to fully expose the entire PCB as done in conventional methods. As a result, this approach offers several advantages, including simplifying the subsequent welding process, and allowing operators to focus on specific areas without concerns about the rest of the sample or the risk of a short circuit. **In the revised manuscript, we have emphasized this novelty of our research in the Discussion section (line 450 – 460).**

Although our work focuses on a specific application (data recovery from MSDs), it has the potential to be extended to various applications across different fields. For example, the proposed robotic-OCT-guided laser ablation technique can be utilized for precise and real-time identification of tumor margins during surgical procedures. Surgeons can accurately target and remove cancerous tissue while minimizing damage to healthy surrounding tissue. In the field of microelectronics, our technique's ability to precisely remove unwanted layers can be valuable for the fabrication of complex integrated circuits, where selective material removal is crucial for optimizing circuit performance. It also opens up new possibilities for broader applications in digital forensics, failure analysis, materials testing, and quality control, bringing advancements in precision and efficiency. **In the revised manuscript, we have emphasized the application scope of our research in the Discussion section (line 473 – 482).**

Overall, our research provides innovative technological and engineering advancements in the inspection and microsurgery of MSDs, resulting in significant contributions to data recovery and reverse engineering. Additionally, our work not only addresses critical challenges in these fields but also holds potential for broader applications beyond our specific focus. Our work has also been recognized for its novelty by the Reviewer #2 and Reviewer #3.

2) *Moreover, some storage devices have a large protective enclosure with a thick layer of plastic. Does the image quality change if a larger SD card has a thicker protective layer?*

Response:

Indeed, the SD card have a large protective enclosure with a thick layer of plastic. However, it is important to note that this plastic casing serves only as an outer protective cover and can be easily removed, as shown in Suppl. Fig. S1(a)(b). When the plastic casing is removed, it can be seen that the essential storage component inside the SD card is still an MSD, as shown in Suppl. Fig. S1(c), allowing robotic-OCT inspection and data recovery to be performed without compromising the integrity of the device. Suppl. Fig. S1(d) shows the OCT image of the internal PCB traces of the MSD retrieved from the SD card.

In the revised manuscript, we have included supplementary material showcasing the inspection results of the SD card. Specifically, in Section 2.7 (line 369 - 371), we have added the following description:

“For an SD card, the essential storage component inside the SD card was still an MSD, and the corresponding robotic-OCT inspection results were shown in Supplementary Figure S1.”

Suppl. Fig. S1. Robotic-OCT inspection of an SD Card. (a) Microscope image of the SD card. (b) Outer plastic casing of the SD card that has been removed. (c) The MSD retrieved from the SD card. (d) The *en face* OCT image showing the internal PCB traces of the MSD on the pinout side.

Reviewer #2 (Remarks to the Author):

General Comments:

The manuscript titled "Robotic-OCT Guided Inspection and Microsurgery of Monolithic Storage Devices" presents a novel approach for non-destructive inspection and microsurgery of monolithic storage devices (MSDs) using a robotic-arm-assisted optical coherence tomography (robotic-OCT) system. The authors describe the capabilities of the system in rapid imaging, defect detection, pin identification, and laser ablation for targeted area removal. The manuscript highlights potential applications in digital forensics, failure analysis, materials testing, and quality control. Overall, the work is of significant novelty and addresses the need for non-destructive techniques in data recovery, and merits the publication. Nevertheless, there are a few points that require further clarification and improvement.

Response:

We thank the Reviewer for the positive comments and recommendation for publication of our manuscript. Please find below the answers to the specific questions.

Specific Comments:

ABSTRACT

The abstract outlines the key features of the robotic-OCT system, including its ability to facilitate high-resolution imaging, PCB visualization, pin identification, defect detection, and selective removal of targeted areas using laser ablation. The abstract also highlights the diverse potential applications of the robotic-OCT technology in digital forensics, failure analysis, materials testing, and quality control. This is an interesting and very practical application topic. However, there are a few suggestions to enhance its clarity and completeness:

- 1) Quantify the benefits: Provide quantitative information on the benefits offered by the robotic-OCT system. For example, mention the achievable imaging resolution, the reduction in inspection time compared to traditional methods, and any improvements in data recovery efficiency observed in the study. Quantitative data will add credibility and demonstrate the practical advantages of the system.*

Response:

Thank you for the reviewer's valuable suggestions. We have revised the abstract to include quantitative information on the benefits offered by the robotic-OCT system, including the achievable imaging resolution, the reduction in inspection time compared to traditional methods, and the improvements in data recovery efficiency observed in the study. All quantitative information in the following revised abstract is highlighted.

"Abstract: Data recovery from monolithic storage devices (MSDs) is in high demand for legal or business purposes. However, the conventional data recovery methods are destructive, complicated, and time-consuming. We develop a robotic-arm-assisted optical coherence tomography (robotic-OCT) for non-destructive inspection of MSDs, offering **~7 μm lateral resolution, ~4 μm axial resolution** and an adjustable field-of-view to accommodate various MSD sizes. Using a continuous scanning strategy, robotic-OCT achieves automated volumetric imaging of a micro-SD card in **~37 seconds**, significantly

faster than the traditional stop-and-stare scanning that typically **takes tens of minutes**. We also demonstrate the robotic-OCT-guided laser ablation as a microsurgical tool for targeted area removal **with precision of $\pm 10 \mu\text{m}$ and accuracy of $\sim 50 \mu\text{m}$** , eliminating the need to remove the entire insulating layer and operator intervention, thus greatly improving the data recovery efficiency. This work has diverse potential applications in digital forensics, failure analysis, materials testing, and quality control."

2) *Expand on the laser ablation technique: Discuss the precision and accuracy of the laser ablation process, potential challenges, and any considerations for ensuring minimal damage to the device during microsurgery.*

Response:

We appreciate the reviewer's valuable suggestions. We have revised the abstract to include the precision and accuracy of the laser ablation process as mentioned in the first comment. In Section 2.6 of the revised manuscript (line 287-335), we have included two additional results to evaluate the precision, accuracy and laser power setting of the laser ablation process.

"First, to assess the accuracy and precision of robotic-OCT guided laser ablation on target areas, we conducted experiments using a total of 18 micro-SD cards of the same model (Card 8). Each card underwent ablation on 10 different technological pins, with each pin considered as an individual trial. To evaluate accuracy, we compared the center position of each ablation hole with that of the corresponding pin in the OCT image. This allowed us to determine how closely the laser ablation process aligned with the intended target. For precision assessment, we compared the center position of each ablation hole with the average center position calculated from the 18 corresponding ablation holes in each trial. This analysis provided insights into the consistency and repeatability of the laser ablation process. As shown in Fig. 6, the precision and accuracy were measured to be $\pm 10 \mu\text{m}$ and $52 \mu\text{m}$ in the X direction, and $\pm 11 \mu\text{m}$ and $50 \mu\text{m}$ in the Y direction. This achieved precision and accuracy level is sufficient for majority of technological pin sizes."

Fig. 6 The accuracy and precision of robotic-OCT guided laser ablation on target areas. (a) Absolute mean error for center positions of the ablation holes in the X (fast-axis) direction, indicating an accuracy of $52 \mu m$ ($n=18$ samples per trial); (b) Center position distribution for the ablation holes in the X (fast-axis) direction, indicating a precision of $\pm 10 \mu m$ ($n=18$ samples per trial); (c) Absolute mean error for center positions of the ablation holes in the Y (slow-axis) direction, indicating an accuracy of $50 \mu m$ ($n=18$ samples per trial); (d) Center position distribution for the ablation holes in the Y (slow-axis) direction, indicating a precision of $\pm 11 \mu m$ ($n=18$ samples per trial). For all box plots, center lines represent the median, the length of the box extends from the lower quartile to the upper quartile, whiskers are 1.5 times of interquartile range and red cross indicates outlier.

“Secondly, we evaluated the influence of different power levels on the laser ablation process, as shown in Fig. 7. We selected a micro-SD card sample (Card 8) contained a 6×6 array of technological pins on its internal PCB, with each pin having a diameter of $\sim 300 \mu m$. Following OCT guidance, laser ablation was performed on each row of pins in the micro-SD card for six power levels: 2 W, 6 W, 10 W, 14 W, 18 W, and 20 W, as shown in Fig. 7(a). By analyzing the *en face* OCT image of the card surface presented in Fig. 7(b), the diameters of the ablation holes at different power levels were measured to be $75 \mu m$, $140 \mu m$, $170 \mu m$, $220 \mu m$, $230 \mu m$, and $260 \mu m$, respectively. In the *en face* OCT image at the depth of $\sim 20 \mu m$ (Fig. 7c), the ablation holes appeared as bright spots and exhibited a progressive increase in size corresponding to the escalating power levels. Regarding ablation depth, the holes formed at the six power levels reached the maximum depths of $9 \mu m$, $12 \mu m$, $18 \mu m$, $20 \mu m$, $22 \mu m$, and $22 \mu m$, respectively, as shown in Fig. 7(d). It was clearly demonstrated that both the size and depth of the ablation holes increased as the laser power level was raised. These results also indicated that the optimal laser power for this specific sample would be 10 W, as it would guarantee that the ablation hole size remained within the pin dimensions, and the ablation depth was close to, but did not exceed, the thickness of the insulating layer to prevent any potential damage to the PCB circuitry. This

observation highlighted the importance of selecting an appropriate laser power level that strikes a balance between achieving the desired ablation results and minimizing excessive heating.”

Fig. 7 Evaluation of different power levels on the laser ablation process. (a) Microscopic image showing a micro-SD card ablated with six different power levels. (b) *En face* OCT image of the ablated card surface, illustrating the diameters of the ablation holes. (c) *En face* OCT image of the ablated card at the depth of $\sim 20 \mu\text{m}$ (pinout side of the PCB), demonstrating the precise positioning and different sizes of the ablation holes. (d) B-scan at the red dashed lines in (b), indicating the maximum depths of the ablation holes at different power levels.

In Discussion section (line 460-469), we have also discussed the potential challenges and considerations for the laser ablation process to ensure minimal damage to the device during microsurgery.

“It is important to note that the precision and accuracy of the laser ablation process determine the minimum pin size that can be accurately and reliably exposed. Our current system achieves accurate and reliable laser ablation for pin sizes above $100 \mu\text{m}$. To further enhance the accuracy and precision, we can employ higher-resolution OCT systems and higher-precision robotic arms that can provide more detailed and precise guidance during the procedure, enabling better visualization of the target area and facilitating improved alignment of the laser beam with the pins. Meanwhile, it is essential to choose a laser power level that aligns with the specific diameter of the pin and the thickness of the insulating layer, which ensures the creation of suitably sized holes to facilitate subsequent processes such as welding and wiring during data recovery, while minimizing any potential damage to the internal circuitry.”

Introduction

The Introduction section provides a comprehensive overview of the challenges and limitations associated with current methods of data recovery from monolithic storage devices (MSDs). It introduces optical coherence tomography (OCT) as a potential solution and highlights its advantages over X-ray radiography. The integration of a robotic arm and the use of continuous scanning are also mentioned as important components of the proposed approach. Overall, the Introduction sets the stage for the study and establishes the motivation for developing the robotic-OCT system. It is a high-quality introduction that deserves praise. However, here are a few suggestions to further improve the Introduction:

- 3) Address potential concerns: Given that OCT is traditionally used in the biomedical field, it may be beneficial to briefly address any potential concerns or challenges in applying OCT to non-biomedical applications such as MSD inspection. This could include factors like sample size, surface reflectivity, or other limitations that might affect the imaging quality or feasibility of the approach.*

Response:

We appreciate the reviewer's praise and suggestion to address potential concerns or challenges in applying OCT to non-biomedical applications such as MSD inspection. In response, we have revised the Introduction section (line 74-86):

"When applying OCT to MSD inspection, several challenges arise, including sample sizes, surface reflectivity, and scanning strategy. Firstly, MSDs come in various types and sizes, ranging from small micro-SD cards to larger MMCs. Conventional OCT systems have a fixed and limited field of view (FOV), making it difficult to adapt to large or complex objects [19]. Moreover, highly reflective surfaces within MSD structures can cause saturation signals and artifacts, hindering the visualization of underlying structures. Optimizing the positions and angles of the OCT scanner can help mitigate direct reflections and improve image quality, but it is challenging to maintain high repeatability and consistency between scans. Furthermore, previous studies often rely on a stop-and-stare scanning strategy to extend the FOV [20-23], where the scanner pauses at specific locations to capture data before moving to the next position. This approach is straightforward and easy to implement. However, it can be time-consuming, inefficient, and result in inconsistent image acquisition due to frequent stopping and repositioning, especially when dealing with larger or irregularly shaped objects."

- 4) Clarify the novelty: Although the introduction mentions that this study is a significant improvement over traditional methods, explicitly emphasizing the novelty of this study can accurately highlight the research value more and contribution of the article and enhance the reader's awareness of the novelty and importance of the article. For example, the need and advantages of combining robotics with optical coherence tomography are emphasized.*

Response:

We appreciate the reviewer's suggestion to explicitly emphasize the novelty of our study

and highlight its research value and contribution. In response, we have revised the Introduction section (line 87-102) to explicitly highlight the innovative aspects of our work. We emphasize the need and advantages of combining robotics with OCT, the implementation of a continuous scanning strategy, and the development of a microsurgical tool for MSDs.

“Combining robotics with OCT offers innovative solutions to overcome these challenges, providing benefits including flexible and accurate imaging, process automation, and consistent image acquisition. The robotic arm can be programmed to move the scanner precisely and automatically to different positions and angles, which allows for an adjustable FOV to accommodate various sample sizes and shapes while reducing saturation signals and artifacts. To overcome the limitations of the traditional stop-and-stare scanning approach with a robotic arm, **a novel continuous scanning strategy** can be implemented, which enables the scanner to acquire data continuously while in motion. By eliminating interruptions or discontinuities in the imaging process, the continuous scanning strategy can significantly reduce the overall scanning time and ensure a uniform and seamless image acquisition across the entire area of interest. Inspired by image-guided surgical interventions in the medical field, **a novel microsurgical tool** can also be developed for MSDs by integrating OCT imaging, robotic arm and laser ablation capabilities. With the aid of OCT images as guidance, the robotic arm can accurately direct the laser to specific locations, enabling selective removal of targeted areas or structures. This approach can minimize damage to the device and facilitate data recovery by eliminating the need to remove the entire insulating layer and reducing operator intervention.”

Methods

The Methods section in this article is well-organized, detailed, and provides a comprehensive overview of the experimental procedures and techniques used to develop the custom-built robotic-OCT system. However, here are a few suggestions to further improve the Introduction:

- 5) *It would be helpful to include more information about the specific components and manufacturers of the OCT engine, such as the brand and model of the super luminescent diode (SLD) and the k-linear spectrometer, OCT probe objective lens size, focal length, working distance. This additional detail will provide a clearer understanding of the setup and equipment used in the study.*

Response:

We appreciate the reviewer’s praise and suggestion to provide more detailed information about the specific components and manufacturers of the OCT engine used in our study.

In the Methods section (line 494-506) of the revised manuscript, we have incorporated the brand and model of the super luminescent diode (SLD), the k-linear spectrometer, OCT probe objective lens size, focal length, and working distance.

“OCT engine: We built a customized spectral-domain OCT engine (Fig. 8a) with $\sim 4\ \mu\text{m}$ axial resolution in air, $\sim 7\ \mu\text{m}$ lateral resolution and 2.27 mm imaging depth. The engine

employed a super luminescent diode (SLD; IPSDW0825, InPhenix) as the light source centered at 850 nm with 105 nm -3dB spectral bandwidth. A custom-built k-linear spectrometer with a F2 prism (PS852, Thorlabs), a 1200 lines/mm diffraction grating (WP-1200/840-25.4, Wasatch Photonics), a fiber-coupled collimator (RC08APC-P01, Thorlabs) and an achromatic lens (#49-381, Edmund) was used to record the linear-in-wavenumber interferogram and eliminate the need for interpolation. The custom-built objective lens of the OCT probe has a diameter of 25.4 mm, a focal length of 25 mm and a working distance of 20 mm. The system's signal-to-noise ratio (SNR) was measured to be 110 dB, with 4.5 mW optical power at the sample and 8.3 μ s integration time for the 2048-pixels 12-bit line-scan CMOS camera (Octoplus, Teledyne e2v, UK), corresponding to 120 kHz A-line rate. We obtained B-scan images by fast-axis lateral scanning, with a duty cycle of 95%, containing 1000 A-lines, resulting in a B-scan frame rate of \sim 114 Hz."

- 6) *It would be valuable to mention any safety measures or precautions taken during the laser ablation microsurgery procedure. This could include details about laser power settings, safety protocols, and any measures implemented to prevent damage to the sample or ensure operator safety.*

Response:

Thank you for your valuable suggestion. In the Methods section (line 576-588) of the revised manuscript, we have included a paragraph specifically dedicated to discussing the safety precautions regarding laser power settings, safety protocols, and measures taken to prevent sample damage and ensure operator safety.

"Safety precautions: First, all personnel operating the laser were required to wear safety glasses specifically designed for the laser's wavelength. This protected their eyes from potential injury caused by laser beams. Additionally, laser power settings were carefully adjusted within safe operating limits to achieve the desired ablation outcome while minimizing any potential risk to the sample or operator. Regular monitoring of the laser power output was conducted to maintain consistent and safe settings. Warning signs were prominently displayed, and unnecessary reflective surfaces were removed from the working area to prevent accidental reflections that could cause harm. Physical barriers and marked safety zones were established around the robotic arm to prevent accidental contact and enable immediate halting of the arm's operation in emergencies. Moreover, the robotic arm's posture was restricted to ensure that the laser beam was directed away from personnel and critical areas. Finally, comprehensive safety training was provided to all individuals involved, covering laser safety protocols, emergency procedures, and safe operation of the equipment."

- 7) *Some graphs, such as Fig. 6 lack scale bars.*

Response:

We appreciate the reviewer's attention to detail and the opportunity to improve the quality of our work. Scale bars have been added to all relevant figures in the revised manuscript to provide a clear indication of the spatial dimensions.

Result

The results section clearly explains the details of the customized robotic-OCT system and the experimental procedure. Here are a few suggestions to further improve the Introduction:

- 8) Clarify the specific benefits and advantages of the continuous scanning strategy. How does this strategy improve the imaging process compared to other scanning methods? Highlighting these advantages and discussing any limitations or trade-offs would provide a more comprehensive evaluation of the approach.*

Response:

We appreciate the reviewer's suggestions. We have included a more detailed discussion on the advantages and limitations of the continuous scanning strategy in the Discussion section (line 425-449) of the revised manuscript.

"Unlike the conventional stop-and-stare scanning approach, our study introduces a novel continuous scanning strategy specifically designed for the robotic-OCT system, where the robotic arm moves continuously over the object and captures data. By eliminating the need for start-stop motions between individual scans, our continuous scanning strategy significantly reduces scanning time and enables large areas to be scanned more efficiently than with conventional methods. For instance, the stop-and-stare approach typically requires ~11 minutes to obtain 1024×256×18 A-lines, including pose optimization and manual scanning point selection operation [22]. In contrast, our continuous scanning strategy is capable of acquiring the same number of A-lines in ~30 seconds, resulting in a remarkable 14-fold increase in scanning speed at the same A-line rate. Moreover, the continuous movement of the robotic arm ensures uniform and seamless image acquisition, significantly eliminating the brightness variations and motion artifacts commonly found in traditional stop-and-stare methods, and enhancing the overall image quality. Furthermore, the strategy can be extended to accommodate larger or irregularly shaped objects by precisely controlling the robotic arm to reposition the scanner at various positions and angles. This adaptability enables customized scanning trajectories based on the unique shape and size of the sample, making the imaging process versatile across a wide range of applications, such as diagnosing and treating organs or tissues, as well as documenting and analyzing artifacts.

Nevertheless, one significant limitation of the continuous scanning strategy is the management of the large amount of data generated during a single scan, especially when imaging larger objects. This can pose challenges in terms of data storage, processing, and analysis. Efficient data processing techniques and robust storage solutions are vital to effectively handle the increased data throughput. Another limitation is the accurate capture of geometric shape and surface details of objects to plan scanning trajectories, particularly for irregularly shaped ones. This limitation can be addressed by incorporating additional imaging technologies, such as a 3D camera, into the robotic-OCT system to provide accurate spatial information."

9) *Address the feasibility of extending the continuous scanning strategy to larger or irregularly shaped objects. Are there any practical limitations or challenges that need to be considered? Discussing the scalability and adaptability of the approach would contribute to the broader applicability of the robotic-OCT system.*

Response:

We appreciate the reviewer's suggestions. Extending the continuous scanning strategy to larger or irregularly shaped objects is feasible and can be an advantage over the conventional scanning methods, but it indeed has practical limitations and challenges that need to be considered. We have discussed the adaptability of the continuous scanning strategy extended to larger or irregularly shaped objects in the Discussion section (line 437-449) of the revised manuscript.

"Furthermore, the strategy can be extended to accommodate larger or irregularly shaped objects by precisely controlling the robotic arm to reposition the scanner at various positions and angles. This adaptability enables customized scanning trajectories based on the unique shape and size of the sample, making the imaging process versatile across a wide range of applications, such as diagnosing and treating organs or tissues, as well as documenting and analyzing artifacts.

Nevertheless, one significant limitation of the continuous scanning strategy is the management of the large amount of data generated during a single scan, especially when imaging larger objects. This can pose challenges in terms of data storage, processing, and analysis. Efficient data processing techniques and robust storage solutions are vital to effectively handle the increased data throughput. Another limitation is the accurate capture of geometric shape and surface details of objects to plan scanning trajectories, particularly for irregularly shaped ones. This limitation can be addressed by incorporating additional imaging technologies, such as a 3D camera, into the robotic-OCT system to provide accurate spatial information."

10) *Additionally, it would be valuable to discuss the potential limitations or challenges encountered during the microsurgery procedure, such as the precision of pin exposure or any thermal effects caused by laser ablation.*

Response:

This is a very important point. Indeed, as suggested by the reviewer, there are two potential challenges encountered during the microsurgery procedure.

One challenge is the precision of pin exposure. Accurate positioning of the robotic arm and alignment of the laser beam with the target area are critical to ensure precise pin exposure. Any deviations or misalignments may result in incomplete or inaccurate ablation, affecting the effectiveness of the procedure. In Section 2.6 of the revised manuscript (line 287-309), we described our experiments to assess the accuracy and precision of robotic-OCT guided laser ablation on target areas.

“First, to assess the accuracy and precision of robotic-OCT guided laser ablation on target areas, we conducted experiments using a total of 18 micro-SD cards of the same model (Card 8). Each card underwent ablation on 10 different technological pins, with each pin considered as an individual trial. To evaluate accuracy, we compared the center position of each ablation hole with that of the corresponding pin in the OCT image. This allowed us to determine how closely the laser ablation process aligned with the intended target. For precision assessment, we compared the center position of each ablation hole with the average center position calculated from the 18 corresponding ablation holes in each trial. This analysis provided insights into the consistency and repeatability of the laser ablation process. As shown in Fig. 6, the precision and accuracy were measured to be $\pm 10 \mu\text{m}$ and $52 \mu\text{m}$ in the X direction, and $\pm 11 \mu\text{m}$ and $50 \mu\text{m}$ in the Y direction. This achieved precision and accuracy level is sufficient for majority of technological pin sizes.”

Another challenge is managing the thermal effects caused by laser ablation. Laser energy can generate heat, which can potentially cause thermal damage to the surrounding materials. It is important to carefully control the laser power output to avoid excessive heating and minimize the risk of thermal damage. In Section 2.6 of the revised manuscript (line 310-335), we described our experiments to evaluate the influence of different power levels on the laser ablation process.

“Secondly, we evaluated the influence of different power levels on the laser ablation process, as shown in Fig. 7. We selected a micro-SD card sample (Card 8) contained a 6×6 array of technological pins on its internal PCB, with each pin having a diameter of $\sim 300 \mu\text{m}$. Following OCT guidance, laser ablation was performed on each row of pins in the micro-SD card for six power levels: 2 W, 6 W, 10 W, 14 W, 18 W, and 20 W, as shown in Fig. 7(a). By analyzing the *en face* OCT image of the card surface presented in Fig. 7(b), the diameters of the ablation holes at different power levels were measured to be $75 \mu\text{m}$, $140 \mu\text{m}$, $170 \mu\text{m}$, $220 \mu\text{m}$, $230 \mu\text{m}$, and $260 \mu\text{m}$, respectively. In the *en face* OCT image at the depth of $\sim 20 \mu\text{m}$ (Fig. 7c), the ablation holes appeared as bright spots and exhibited a progressive increase in size corresponding to the escalating power levels. Regarding ablation depth, the holes formed at the six power levels reached the maximum depths of $9 \mu\text{m}$, $12 \mu\text{m}$, $18 \mu\text{m}$, $20 \mu\text{m}$, $22 \mu\text{m}$, and $22 \mu\text{m}$, respectively, as shown in Fig. 7(d). It was clearly demonstrated that both the size and depth of the ablation holes increased as the laser power level was raised. These results also indicated that the optimal laser power for this specific sample would be 10 W, as it would guarantee that the ablation hole size remained within the pin dimensions, and the ablation depth was close to, but did not exceed, the thickness of the insulating layer to prevent any potential damage to the PCB circuitry. This observation highlighted the importance of selecting an appropriate laser power level that strikes a balance between achieving the desired ablation results and minimizing excessive heating.”

In the Discussion section (line 460-469), we have also discussed the potential considerations for the laser ablation process to ensure minimal damage to the device during microsurgery.

"It is important to note that the precision and accuracy of the laser ablation process determine the minimum pin size that can be accurately and reliably exposed. Our current system achieves accurate and reliable laser ablation for pin sizes above 100 μm . To further enhance the accuracy and precision, we can employ higher-resolution OCT systems and higher-precision robotic arms that can provide more detailed and precise guidance during the procedure, enabling better visualization of the target area and facilitating improved alignment of the laser beam with the pins. Meanwhile, it is essential to choose a laser power level that aligns with the specific diameter of the pin and the thickness of the insulating layer, which ensures the creation of suitably sized holes to facilitate subsequent processes such as welding and wiring during data recovery, while minimizing any potential damage to the internal circuitry."

11) *Discussion*: The authors' discussion highlights the novelty and potential impact of their work on the field of data recovery from microsurgery for MSDs. They provide a clear contrast between the traditional methods of data recovery, which are time-consuming, require expertise, and can cause damage to the samples, and the proposed robotic-OCT system, which enables automated, non-destructive, and fast imaging. The authors rightly point out that their work has the potential to revolutionize data recovery procedures for MSDs.

Response:

Thank you for the reviewer's comments on the discussion section of our work. We appreciate the reviewer's recognition of the novelty and potential impact of our robotic-OCT system on data recovery from microsurgery for MSDs.

Reviewer #3 (Remarks to the Author):

The authors do a good job introducing the optical coherence tomography as a safe and fast option for inspecting the internal PCB layers of the monolith flash storage devices which can substitute the traditional X-ray inspection. The authors also automated the whole process which can help make forensic data extraction more efficient than before.

This research can be a good reference for digital forensics labs to introduce OCT instead of X-ray for reverse-engineering the target device PCBs. Also, this work can contribute to make the data acquisition process quicker than before in digital forensics.

Response:

We would like to express our sincere appreciation to the reviewer for the recognition of our research and highlighting the impact of our work, as well as providing valuable suggestions to improve it. Please find below the answers to the specific questions.

- 1) *However, one thing I feel missing is the detailed evaluation of the OCT inspection. The authors do mention the limitation saying that the 2nd layer of the PCB cannot be observed due to the thickness of the epoxy. But up to how many micro meters can the OCT penetrate to observe further layers? If there are more than 2 layers of PCBs, then how many layers can be inspected? And does the material of the epoxy affect the efficiency of OCT inspection? Those details should be discussed in the article.*

Response:

We appreciate the reviewer's comments regarding the detailed evaluation of the OCT inspection in our article. We would like to address the reviewer's concerns and provide further clarification.

"The authors do mention the limitation saying that the 2nd layer of the PCB cannot be observed due to the thickness of the epoxy. But up to how many micro meters can the OCT penetrate to observe further layers?"

In the Discussion section of the original manuscript, we mentioned, "one of the primary limitations of OCT is the relatively low penetration depth, which makes it difficult to obtain an image of the PCB traces on the dice side because the plastic substrate is too thick." We apologize for the misunderstanding of this sentence and would like to clarify and provide further explanation regarding this statement.

As illustrated in Fig. 2(c), the micro-SD card structure comprises a PCB hosting electronic components, a protective plastic housing on the dice side of the PCB, and an insulating layer on the pinout side of the PCB. In our study, when imaging from the pinout side, we can only visualize a portion of the 2nd layer (dice side) of the PCB. This limitation is primarily attributed to the high reflectivity of the copper layer on the pinout side, rather than the thickness of the epoxy. Attempting to image the card from the dice side is not effective due to the light penetration being hindered by the thick plastic substrate. Consequently, no useful information can be obtained from this side. Since the majority of pertinent information for data recovery is concentrated on the pinout side, imaging the card from this side represents an optimized approach for our specific application.

To assess the penetration depth of our OCT system into the micro-SD card, we conducted a quantitative evaluation as described in Section 2.2 of the revised manuscript (line 161-181).

“In Fig. 2(e), we presented averaged A-scans (n=10) obtained from two distinct positions, labeled as P1 and P2. These positions were identified and marked using yellow rectangles in Fig. 2(d). By analyzing the A-scans, we made several important observations. First, both A-scans exhibited a clear first peak (S1), which corresponded to the surface of the micro-SD card, indicating the starting point of the penetration depth. Moving further into the card, the A-scan from position P1 displayed a second peak (S2), representing the boundary between the insulating layer and the PCB. This peak resulted from the reflection of the incident light by the copper conductive layer on the pinout side of the PCB. By calculating the optical path length (OPL) between S1 and S2, we estimated the thickness of the insulating layer to be **~23 μm** (See Methods section for details). In addition, the A-scan from position P2 exhibited a second peak (S3), indicating the reflection of light from the copper conductive layer on the dice side of the PCB. Notably, there was no corresponding copper conductive layer at the same position on the pinout side, as evident from the absence of a reflected signal. By calculating the OPL between S2 and S3, we estimated the thickness of the PCB to be **~250 μm** . Consequently, the total penetration depth achieved by our OCT system into this sample was determined to be **~273 μm** .”

Fig. 2. Tomographic analysis of the multilayer structure of a micro-SD Card. 3D OCT images from (a) top-to-bottom view and (b) cross-sectional view. (c) The diagram of the multilayer structure of a micro-SD card. (d) The *en face* OCT images at three different depths. Top: surface (Depth = 0 μm). Middle: PCB pinout side (Depth = 23 μm). Bottom: PCB dice side (Depth = 273 μm). (e) The averaged A-scans at two different positions P1 and P2 marked as yellow rectangles in (d).

"If there are more than 2 layers of PCBs, then how many layers can be inspected?"

In the case of MSDs, as illustrated in Fig. 2(c), they usually only consist of two layers: the pinout side and the dice side. It is uncommon to have additional layers beyond these two. Even if there were additional layers, the ability to inspect information from deeper layers would be further hindered due to the reflectivity of the copper layer. However, it is essential to emphasize that the majority of crucial information required for data recovery is typically concentrated within the first layer (pinout side). Therefore, the inability to observe deeper layers has minimal impact on the data recovery process.

"And does the material of the epoxy affect the efficiency of OCT inspection?"

The material used for the black insulating layer on the pinout side of micro-SD cards may vary among different manufacturers or even different product versions. However, we conducted OCT imaging on a diverse set of over 80 micro-SD cards, including different brands and models collected from multiple sources (see Supplementary Figure S2). Our observations consistently revealed that the insulating layers of these cards exhibited transparency to the near-infrared light used in our OCT systems. In Fig. 3, we presented representative OCT imaging results of four micro-SD cards, each with distinct brands, models, surface roughness, and insulating layer thicknesses. The B-scans clearly showed the presence of a black region within the insulating layer, indicating that no light was scattered from the inside of the layer. These results demonstrated that the material of the insulating layer did not significantly affect the OCT inspection process, as it was transparent to near-infrared light. We have incorporated these results into the Section 2.3 of the revised manuscript (line 193-199).

- 2) *The authors suggest using the mid-infrared OCT, but no further detail is discussed. Please make it clear what becomes better and what limitation the operators would face by changing the wavelength of the infrared.*

Response:

We appreciate the reviewer's suggestion. Mid-infrared OCT [29][30] can be employed to reduce scattering effects and improve penetration through the thicker plastic substrate on the dice side. This would enable better visualization of PCB traces on the dice side. However, it introduces another challenge that the electronic components and copper layers on the dice side of the PCB may impede mid-infrared light penetration, affecting the visibility of PCB traces on the pinout side. For effective visualization of PCB traces on both sides, it is favorable to use mid-infrared OCT separately for each side.

Meanwhile, it is crucial to consider the limitations associated with mid-infrared OCT to ensure its effective utilization for MSD inspection. One significant limitation is the potential degradation of spatial resolution compared to near-infrared OCT. The longer wavelength used in mid-infrared OCT results in reduced axial and lateral resolution, which can impact the ability to visualize fine details and subtle features within the MSD, potentially affecting the accuracy of inspection and data recovery processes. Additionally, mid-infrared OCT often faces challenges with lower signal-to-noise ratio (SNR) due to the scarcity of ideal mid-infrared detectors, leading to reduced image quality. Another limitation is the increased complexity and costs associated with implementing mid-infrared OCT, as it requires specialized components and system apparatus compared to traditional near-infrared OCT systems.

We have added a discussion on the potential improvements and limitations that operators may encounter by changing the wavelength to the mid-infrared region in the Discussion section of the revised manuscript (line 408-416).

- 3) *Overall, the article lacks detailed explanation of each steps and investigations. For example, in section 2.5, the authors inspect variable damaged SD cards, but its results*

are pretty vague. For example, for the scratched card, how deep is the scratch based on the OCT inspection? Is it reaching the flash die and the data is not recoverable? Other sections are also a bit too much summarized. Please try to be more descriptive. The work itself is nice and helpful for forensic investigation. Since applying OCT in device inspection is the main part of this work (in addition to automating the laser ablation process), the readers would expect more evaluation details.

Response:

Thank you for your valuable suggestions. We agree that the work is helpful for forensic investigation, and providing more detailed explanations of each step would be helpful for enhancing the clarity and comprehensibility of the work. We have incorporated more explanations and discussions in the revised manuscript to address your concerns and have updated Fig. 2 and Fig. 5 accordingly.

Fig. 5 OCT images of the scratched (Card 5), cracked (Card 6) and burned (Card 7) micro-SD card. The *en face* OCT images of the PCB traces on the pinout side of (a) Card 5, (e) Card 6, (i) Card 7. The enlarged *en face* OCT images of the corresponding areas (indicated by the red boxes) of (b) Card 5, (f) Card 6, (j) Card 7; The microscopic images of (c) Card 5, (g) Card 6, (k) Card 7, and the corresponding B-scans (indicated by the red dashed lines) of (d) Card 5, (h) Card 6, (l) Card 7.

In Section 2.5, we have made improvements to the inspection of **the scratched SD card**. We now provide measurements of the depths of both minor and severe scratches using B-scans in the revised manuscript (line 256-262). The minor scratch was found to have a depth of $\sim 14 \mu\text{m}$, which is smaller than the thickness of the insulating layer ($\sim 30 \mu\text{m}$). This indicated that the scratch did not cause any damage to the pinout side of the PCB. On the other hand, the severe scratch had a depth of $\sim 60 \mu\text{m}$, exceeding the thickness of the insulating layer and suggesting a potential risk of damaging the pinout side of the PCB. However, considering the overall PCB thickness of $\sim 250 \mu\text{m}$, we can confidently conclude that the scratch did not reach the dice side and did not impact the flash die. These additional details provided a more precise understanding of the scratches and their effects on the micro-SD card.

Additionally, we have included further descriptions regarding cracked and burnt cards. **Regarding the cracked card** (line 264-267), although it rendered the micro-SD card unusable, our examination of *en face* OCT images revealed that the fractures did not affect the core pins necessary for data recovery. This finding indicated that there was still a possibility that data recovery procedures could be performed on the intact region. **Regarding the burnt card** (line 270-272), we could identify the burnt areas through the B-scans, as illustrated by the red boxed area in Fig. 5(i). By thoroughly analyzing all the B-scans, we could delineate the boundary of the burnt area in the *en face* OCT image as shown in Fig. 5(j). These results suggested that our method could effectively detect and evaluate a variety of defects such as the cutting traces, scratches, cracked traces, burned areas and broken connections, for repairing or recovering data from the damaged devices, which could help determine whether the card was repairable or needed to be replaced, and provide valuable information for the following data recovery.

Furthermore, we've provided more detailed descriptions in other sections highlighted in yellow.

- a) **In Section 2.1**, we have included details such as the speed of the robotic arm and the data acquisition time for each step.
- b) **In Section 2.2**, we have included the thickness of the insulating layer ($\sim 23 \mu\text{m}$) and the PCB ($\sim 250 \mu\text{m}$), as well as the overall penetration depth ($\sim 273 \mu\text{m}$). These important details have been marked on Fig. 2.
- c) **In Section 2.3**, we have added a brief description of the four different micro-SD card samples and included the corresponding measurement values in the text.
- d) **In Section 2.4**, we have provided an explanation for the selection of a depth of $\sim 66 \mu\text{m}$ for the *en face* OCT imaging. The choice of imaging at this specific depth was made to align with the middle portion of the PCB, effectively avoiding potential interference from PCB wiring and solid soldering holes, and ensuring clear visualization of the vias.
- e) **In Section 2.6**, we have included two additional results to evaluate the laser ablation process.

First, to assess the accuracy and precision of robotic-OCT guided laser ablation on

target areas, we conducted experiments using a total of 18 micro-SD cards of the same model (Card 8). Each card underwent ablation on 10 different technological pins, with each pin considered as an individual trial. To evaluate accuracy, we compared the center position of each ablation hole with that of the corresponding pin in the OCT image. This allowed us to determine how closely the laser ablation process aligned with the intended target. For precision assessment, we compared the center position of each ablation hole with the average center position calculated from the 18 corresponding ablation holes in each trial. This analysis provided insights into the consistency and repeatability of the laser ablation process. As shown in Fig. 6, the precision and accuracy were measured to be $\pm 10 \mu\text{m}$ and $52 \mu\text{m}$ in the X direction, and $\pm 11 \mu\text{m}$ and $50 \mu\text{m}$ in the Y direction. This achieved precision and accuracy level is sufficient for majority of technological pin sizes.

Secondly, we evaluated the influence of different power levels on the laser ablation process, as shown in Fig. 7. We selected a micro-SD card sample (Card 8) contained a 6×6 array of technological pins on its internal PCB, with each pin having a diameter of $\sim 300 \mu\text{m}$. Following OCT guidance, laser ablation was performed on each row of pins in the micro-SD card for six power levels: 2 W, 6 W, 10 W, 14 W, 18 W, and 20 W, as shown in Fig. 7(a). By analyzing the *en face* OCT image of the card surface presented in Fig. 7(b), the diameters of the ablation holes at different power levels were measured to be $75 \mu\text{m}$, $140 \mu\text{m}$, $170 \mu\text{m}$, $220 \mu\text{m}$, $230 \mu\text{m}$, and $260 \mu\text{m}$, respectively. In the *en face* OCT image at the depth of $\sim 20 \mu\text{m}$ (Fig. 7c), the ablation holes appeared as bright spots and exhibited a progressive increase in size corresponding to the escalating power levels. Regarding ablation depth, the holes formed at the six power levels reached the maximum depths of $9 \mu\text{m}$, $12 \mu\text{m}$, $18 \mu\text{m}$, $20 \mu\text{m}$, $22 \mu\text{m}$, and $22 \mu\text{m}$, respectively, as shown in Fig. 7(d). It was clearly demonstrated that both the size and depth of the ablation holes increased as the laser power level was raised. These results also indicated that the optimal laser power for this specific sample would be 10 W, as it would guarantee that the ablation hole size remained within the pin dimensions, and the ablation depth was close to, but did not exceed, the thickness of the insulating layer to prevent any potential damage to the PCB circuitry. This observation highlighted the importance of selecting an appropriate laser power level that strikes a balance between achieving the desired ablation results and minimizing excessive heating.

Fig. 6. The accuracy and precision of robotic-OCT guided laser ablation on target areas. (a) Absolute mean error for center positions of the ablation holes in the X (fast-axis) direction, indicating an accuracy of $52 \mu\text{m}$ ($n=18$ samples per trial); (b) Center position distribution for the ablation holes in the X (fast-axis) direction, indicating a precision of $\pm 10 \mu\text{m}$ ($n=18$ samples per trial); (c) Absolute mean error for center positions of the ablation holes in the Y (slow-axis) direction, indicating an accuracy of $50 \mu\text{m}$ ($n=18$ samples per trial); (d) Center position distribution for the ablation holes in the Y (slow-axis) direction, indicating a precision of $\pm 11 \mu\text{m}$ ($n=18$ samples per trial). For all box plots, center lines represent the median, the length of the box extends from the lower quartile to the upper quartile, whiskers are 1.5 times of interquartile range and red cross indicates outlier.

Fig. 7 Evaluation of different power levels on the laser ablation process. (a) Microscopic image showing a micro-SD card ablated with six different power levels. (b) *En face* OCT image of the ablated card surface, illustrating the diameters of the ablation holes. (c) *En face* OCT image of the ablated card at the depth of $\sim 20 \mu\text{m}$ (pinout side of the PCB), demonstrating the precise positioning and different sizes of the ablation holes. (d) B-scan at the red dashed lines in (b), indicating the maximum depths of the ablation holes at different power levels.

4) Also please reconsider the structure of the article. The whole time I was missing the background information (the one mentioned in section 4) and it was hard to follow all the sections.

Response:

We appreciate the reviewer’s concern regarding the background information and its placement in the manuscript. We would like to emphasize that in accordance with the formatting guidelines of Nature Communications, the detailed background information and methodology are indeed intended to be presented in Section 4. However, in order to make it easier for readers to follow the content, we have added indications (“see Methods section for details”) throughout the article to direct readers to Section 4 for more detailed background information and methodology.

Reviewer #2 (Remarks to the Author):

I would like to express my appreciation for the modifications the authors have made in response to my previous suggestions and comments. Overall, the modifications made in response to my suggestions have enriched the content of the manuscript, presenting the significance and novelty of the research more clearly. The meticulous revisions and comprehensive responses have significantly improved the manuscript, making it more refined and complete. Considering the thorough responses to all the review comments and suggestions, I believe the manuscript is now ready for publication in Nature Communications.

Reviewer #3 (Remarks to the Author):

The authors have addressed all my questions and concerns well. The overall novelty of this research is still somewhat limited since the authors focus only on physical non-invasive inspection of the damaged microSDs (and USB thumbdrives) where they already have the database of the technical pins as mentioned in line 226. When performing the actual data recovery (which seems to be out of the scope of this paper but obviously the very important next step), often times forensic investigators do not know those technological pin assignment, and those pins for accessing the internal flash memory needs to be identified first. It can be done by tracing the PCB traces from where the bonding wires of the flash memory die are connected. From what I understand from the article, OCT cannot trace the bonding wires to identify those pins? Therefore forensic investigators still need to use X-ray or other systems to identify those technological pins when performing the actual data recovery.

Nevertheless, this article can be a good reference for the forensic community when they consider acquiring an OCT system to a digital forensic lab.

Point-by-point response to the Reviewers' comments

We thank the Reviewers for taking the time to review our manuscript again and for providing further valuable comments and suggestions. We have revised the manuscript to address the remaining concerns of the reviewers.

REVIEWER COMMENTS

Reviewer #2 (Remarks to the Author):

I would like to express my appreciation for the modifications the authors have made in response to my previous suggestions and comments. Overall, the modifications made in response to my suggestions have enriched the content of the manuscript, presenting the significance and novelty of the research more clearly. The meticulous revisions and comprehensive responses have significantly improved the manuscript, making it more refined and complete. Considering the thorough responses to all the review comments and suggestions, I believe the manuscript is now ready for publication in Nature Communications.

Response:

We thank the Reviewer for recognizing the efforts we have made in addressing the previous suggestions and comments. We deeply appreciate the Reviewer's positive comments and recommendation for the publication of our manuscript.

Reviewer #3 (Remarks to the Author):

The authors have addressed all my questions and concerns well. The overall novelty of this research is still somewhat limited since the authors focus only on physical non-invasive inspection of the damaged microSDs (and USB thumbdrives) where they already have the database of the technical pins as mentioned in line 226. When performing the actual data recovery (which seems to be out of the scope of this paper but obviously the very important next step), often times forensic investigators do not know those technological pin assignment, and those pins for accessing the internal flash memory needs to be identified first. It can be done by tracing the PCB traces from where the bonding wires of the flash memory die are connected. From what I understand from the article, OCT cannot trace the bonding wires to identify those pins? Therefore forensic investigators still need to use X-ray or other systems to identify those technological pins when performing the actual data recovery. Nevertheless, this article can be a good reference for the forensic community when they consider acquiring an OCT system to a digital forensic lab.

Response:

We appreciate the Reviewer's recognition of our work as a good reference for the forensic community, as well as the insightful comments and feedback on our manuscript. We would like to address the Reviewer's concerns regarding the significance of our proposed method, its potential impact on the data recovery process, and the utilization of OCT for identifying technological pins.

In practical applications, the data recovery process for monolithic storage devices (MSDs) can be generally divided into two main steps. The first step involves sample treatment, which encompasses a series of tasks such as completely removing the insulating layers, precisely identifying the technological pins, and meticulously welding these pins to the adapter. The subsequent step is data extraction, referred to as "the actual data recovery" by the Reviewer, which is a standardized procedure that typically relies on an automated and commercially available data recovery system. In these two steps, the sample treatment phase emerges as the most challenging aspect of the entire data recovery process, primarily due to its labor-intensive and time-consuming nature. Therefore, this problem serves as the primary focus of this study.

In this study, our proposed method not only focuses on physical non-invasive inspection of the MSDs, but also performs precise microsurgery for MSDs by accurately and automatically removing unwanted layers or structures while minimizing sample damage. By doing so, it selectively exposes the relevant pins necessary for data recovery from the flash memory, eliminating the need to fully expose the entire PCB as done in conventional methods, simplifying the subsequent welding process, and allowing operators to focus on specific areas without concerns about the rest of the sample or the risk of a short circuit. Consequently, our method represents a significant advancement in enhancing data recovery efficiency by revolutionizing traditional sample treatment procedures.

While we acknowledge that OCT cannot trace the bonding wires of the flash memory die, it can effectively capture the image of the PCB layout on the pinout side of the MSD. This image can then be matched with entries in our current PCB layout database. If a matching layout is found, it signifies the presence of that specific MSD model in the database, enabling us to directly identify the relevant technological pins. However, if the MSD model is not present in the database, the pin assignments can still be determined by selectively exposing all the technological pins using robotic-OCT-guided laser ablation microsurgery and connecting each exposed pin to a logic analyzer for function analysis. This procedure eliminates the need for completely removing the insulating layer and ensures minimal damage to the MSD sample.

In Discussion section (line 429-437) highlighted in yellow, we have discussed the utilization of OCT for identifying technological pins.

"While OCT cannot trace the bonding wires of the flash memory die, it can effectively capture the image of the PCB layout on the pinout side of the MSD. This image can then be matched with entries in our current PCB layout database. If a matching layout is found, it signifies the presence of that specific MSD model in the database, enabling us to directly identify the relevant technological pins. However, if the MSD model is not present in the database, the pin assignments can still be determined by selectively exposing all the technological pins using robotic-OCT-guided laser ablation microsurgery and connecting each exposed pin to a logic analyzer for function analysis. This procedure eliminates the need for completely removing the insulating layer and ensures minimal damage to the MSD sample."